# Overall energy conversion efficiency of a photosynthetic vesicle

**Melih Sener[1,2]\*, Johan Strumpfer [1,3], Abhishek Singharoy[1], C Neil Hunter[4], Klaus Schulten[1,2,3]\***

[1]Beckman Institute for Advanced Science and Technology, University of Illinois at Urbana-Champaign, Urbana, United States; [2]Department of Physics, University of Illinois at Urbana-Champaign, Urbana, United States; [3]Center for Biophysics and Computational Biology, University of Illinois at Urbana-Champaign, Urbana, United States; [4]Department of Molecular Biology and Biotechnology, University of Sheffield, Sheffield, United Kingdom

**Abstract** The chromatophore of purple bacteria is an intracellular spherical vesicle that exists in numerous copies in the cell and that efficiently converts sunlight into ATP synthesis, operating typically under low light conditions. Building on an atomic-level structural model of a low-light-adapted chromatophore vesicle from *Rhodobacter sphaeroides*, we investigate the cooperation between more than a hundred protein complexes in the vesicle. The steady-state ATP production rate as a function of incident light intensity is determined after identifying quinol turnover at the cytochrome $bc_1$ complex (cyt$bc_1$) as rate limiting and assuming that the quinone/quinol pool of about 900 molecules acts in a quasi-stationary state. For an illumination condition equivalent to 1% of full sunlight, the vesicle exhibits an ATP production rate of 82 ATP molecules/s. The energy conversion efficiency of ATP synthesis at illuminations corresponding to 1%–5% of full sunlight is calculated to be 0.12–0.04, respectively. The vesicle stoichiometry, evolutionarily adapted to the low light intensities in the habitat of purple bacteria, is suboptimal for steady-state ATP turnover for the benefit of protection against over-illumination.

\*For correspondence: melih@ks. uiuc.edu (MS); kschulte@ks.uiuc. edu (KS)

**Competing interests:** The authors declare that no competing interests exist.

## Introduction

Energy for most life on Earth is provided by sunlight harvested by photosynthetic organisms, which have evolved a wide variety of mechanisms for utilizing light energy to drive cellular processes (*Blankenship, 2014*). These organisms absorb sunlight and subsequently utilize the Förster mechanism (*Sener et al., 2011*) and quantum coherence (*Strümpfer et al., 2012*; *Panitchayangkoon et al., 2010*; *Scholes, 2010*) for efficient excitation energy transfer, followed by conversion of light energy into chemical energy (*Feniouk and Junge, 2009*). The light harvesting system of purple bacteria (*Hu et al., 2002*; *Cartron et al., 2014*) is claimed to be the earliest of the current photosynthetic lineages (*Xiong et al., 2000*) and exhibits, at the supra-molecular level as well as at the level of individual proteins, less complexity than the thylakoid membranes of the more ubiquitous cyanobacteria and plants (*Kirchhoff et al., 2002*).

In the purple bacterium *Rhodobacter (Rba.) sphaeroides* the basic photosynthetic unit is the chromatophore (*Cogdell et al., 2006*; *Strümpfer et al., 2011*; *Cartron et al., 2014*), a 50–70 nm diameter vesicle, as shown in *Figure 1*, formed through invagination of the intracytoplasmic membrane (*Tucker et al., 2010*; *Gubellini et al., 2007*) and comprising over a hundred protein complexes (*Jackson et al., 2012*; *Woronowicz and Niederman, 2010*; *Woronowicz et al., 2013*). The proteins that constitute the chromatophore are primarily the light harvesting (LH) complexes, photosynthetic reaction centers (RCs), cyt$bc_1$ complexes, and ATP synthases, which cooperate to harvest light

**eLife digest** Photosynthesis, or the conversion of light energy into chemical energy, is a process that powers almost all life on Earth. Plants and certain bacteria share similar processes to perform photosynthesis, though the purple bacterium *Rhodobacter sphaeroides* uses a photosynthetic system that is much less complex than that in plants. Light harvesting inside the bacterium takes place in up to hundreds of compartments called chromatophores. Each chromatophore in turn contains hundreds of cooperating proteins that together absorb the energy of sunlight and convert and store it in molecules of ATP, the universal energy currency of all cells.

The chromatophore of primitive purple bacteria provides a model for more complex photosynthetic systems in plants. Though researchers had characterized its individual components over the years, less was known about the overall architecture of the chromatophore and how its many components work together to harvest light energy efficiently and robustly. This knowledge would provide insight into the evolutionary pressures that shaped the chromatophore and its ability to work efficiently at different light intensities.

Sener et al. now present a highly detailed structural model of the chromatophore of purple bacteria based on the findings of earlier studies. The model features the position of every atom of the constituent proteins and is used to examine how energy is transferred and converted. Sener et al. describe the sequence of energy conversion steps and calculate the overall energy conversion efficiency, namely how much of the light energy arriving at the microorganism is stored as ATP.

These calculations show that the chromatophore is optimized to produce chemical energy at low light levels typical of purple bacterial habitats, and dissipate excess energy to avoid being damaged under brighter light. The chromatophore's architecture also displays robustness against perturbations of its components. In the future, the approach used by Sener et al. to describe light harvesting in this bacterial compartment can be applied to more complex systems, such as those in plants.

energy for photophosphorylation. The architecture of the chromatophore, reported in (*Şener et al., 2007, 2010; Cartron et al., 2014*), has been determined by combining atomic force microscopy (AFM) (*Bahatyrova et al., 2004; Olsen et al., 2008*), cryo-electron microscopy (cryo-EM) (*Qian et al., 2005; Cartron et al., 2014*), crystallography (*Koepke et al., 1996; McDermott et al., 1995; Papiz et al., 2003; Jamieson et al., 2002*), optical spectroscopy (*Hunter et al., 1985; Şener et al., 2010*), mass spectroscopy (*Cartron et al., 2014*), and proteomics (*Jackson et al., 2012; Woronowicz and Niederman, 2010; Woronowicz et al., 2013*) data. The composition of the chromatophore depends on growth conditions such as light intensity (*Adams and Hunter, 2012; Woronowicz et al., 2011b, 2011a*) and can also be influenced by mutations (*Siebert et al., 2004; Hsin et al., 2010b*).

The chromatophore displays organizational principles for the integration of multiple processes (*Şener et al., 2010*). Evolutionary competition at the organism level has driven photosynthetic sub-systems toward optimal and robust function (*Noy et al., 2006; Noy, 2008; Scholes et al., 2011*) that can guide the design of artificial light harvesting devices such as biohybrid antennas (*Harris et al., 2013*) and nanopatterned light harvesting (LH) complex arrays (*Reynolds et al., 2007; Vasilev et al., 2014; Patole et al., 2015*). The development and improvement of such artificial, biological, or biohybrid light harvesting systems may alleviate mankind's future energy demand (*Blankenship et al., 2011*).

The functional principles displayed by the chromatophore and prevalent also in other photosynthetic systems include efficient excitonic coupling between components (*Hu et al., 1997, 1998; van Grondelle and Novoderezhkin, 2006b; Olaya-Castro et al., 2008; Şener et al., 2011*), the utilization of quantum coherence (*Ishizaki and Fleming, 2009a; Strümpfer et al., 2012*), photo-protection by carotenoids (*Damjanović et al., 1999*), accommodation of thermal fluctuations, studied through experimental (*Visscher et al., 1989; van Grondelle et al., 1994; Pullerits et al., 1994; Gobets et al., 2001; Janusonis et al., 2008; Freiberg et al., 2009*) as well as theoretical (*Damjanović et al., 2002; Şener and Schulten, 2002; Ishizaki and Fleming, 2009b; van Grondelle*

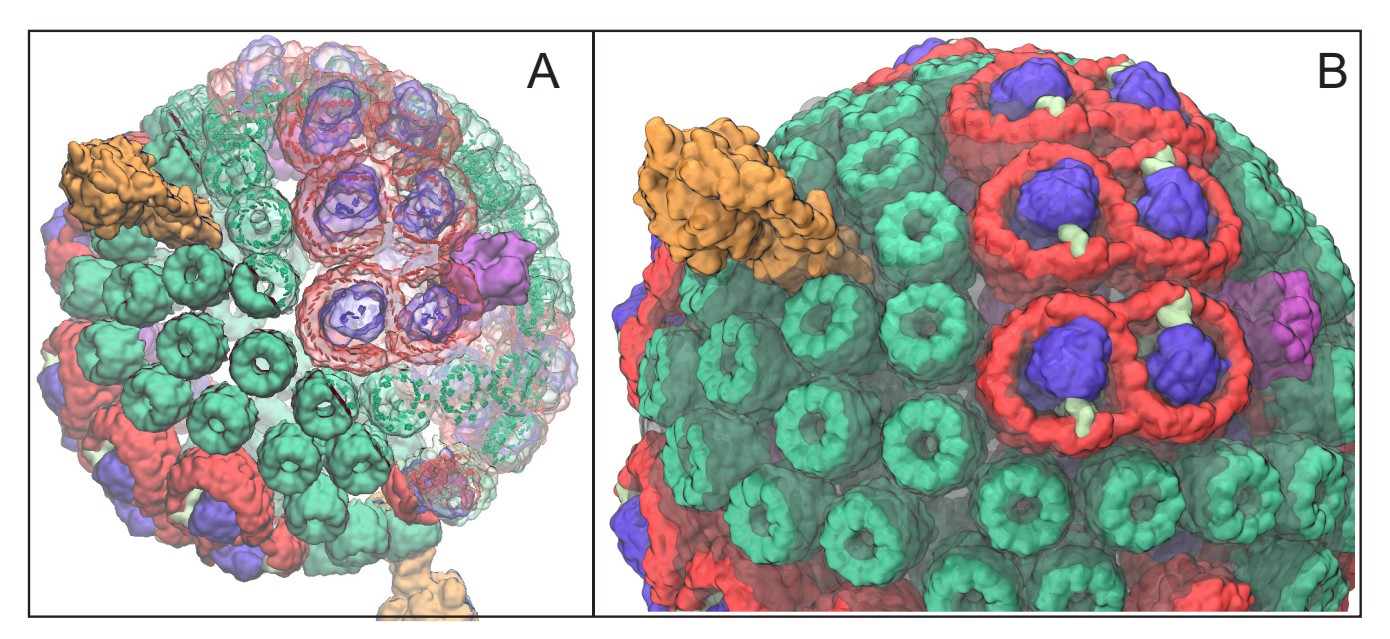

**Figure 1.** Atomic structural model of a low-light-adapted chromatophore vesicle from *Rba. sphaeroides*. The model is based on AFM, EM, crystallography, mass spectroscopy, proteomics, and optical spectroscopy data (*Cartron et al., 2014*). The inner diameter of the vesicle is 50 nm. The model considered in this study is a variant of the one reported in (*Cartron et al., 2014*), which features 63 LH2 complexes (green), 11 dimeric and 2 monomeric RC-LH1-PufX complexes (LH1:red; RC:blue; PufX:lime), 4 cyt$bc_1$ (magenta), and 2 ATP synthases (orange), as well as 2469 BChls and 1542 carotenoids. Proteins are shown in surface representation. (**A**) Proteins and BChls of the chromatophore. Some of the light harvesting proteins are rendered transparent to reveal their BChl pigments. BChls are represented by their porphyrin rings only. See *Video 1* presenting the vesicle. (**B**) Close-up of chromatophore showing its lipid membrane (transparent) along with its proteins colored as in (**A**). The membrane of 16,000 lipids contains the quinone/quinol pool of about 900 molecules. Energy conversion in the chromatophore proceeds in three stages: (I) light harvesting and electron transfer reducing the quinone pool; (II) quinone/quinol diffusion and exchange of quinols for quinones at cyt$bc_1$ (thereby generating a proton gradient across the vesicle membrane) as well as diffusive motion of cytochrome $c_2$ inside the chromatophore shuttling single electrons from cyt$bc_1$ to RC; (III) utilization of the proton gradient for ATP synthesis.

*and Novoderezhkin, 2006b*; *Strümpfer and Schulten, 2011*, *2012a*) methods. The chromatophore exhibits also the features of modularity, repair, and assembly of components (*Hsin et al., 2010a*), high quantum yield of organelle-scale pigment networks (*Şener et al., 2007*, *2010*; *Cartron et al., 2014*), isolation of the electron transfer chains (*Şener and Schulten, 2008*), co-accommodation of competing functions such as efficient energy transfer and diffusion in the quinone/quinol pool (*Lavergne et al., 2009*; *Sener et al., 2010*), as well as adaptation to changing external conditions (*Adams and Hunter, 2012*; *Woronowicz et al., 2011a*; *Niederman, 2013*; *Woronowicz et al., 2013*).

Energy conversion in the chromatophore proceeds in three stages as discussed below: (i) light harvesting and charge separation, converting quinone into quinol at a RC; (ii) diffusion of quinone/quinol in the chromatophore membrane and cytochrome $c_2$ diffusion inside the chromatophore vesicle, resulting, at cyt$bc_1$, in the generation of a proton gradient as well as a transmembrane electrochemical gradient across the chromatophore membrane (henceforth referred to collectively as proton gradient); (iii) utilization of the proton gradient, culminating in ADP binding and ATP release at ATP synthase. The quinone/quinol as well as the generated proton-motive force function as energy buffers between light harvesting and ATP synthesis stages. The proton gradient along with the redox states of the quinone/quinol pool are influenced by the enzymes succinate dehydrogenase, NADH dehydrogenase, cytochrome $c$ oxidase, and ubiquinol oxidase (*Bowyer et al., 1985*; *Klamt et al., 2008*). A summary for the energy conversion processes in the chromatophore can be found in (*Klamt et al., 2008*; *Sener et al., 2014*).

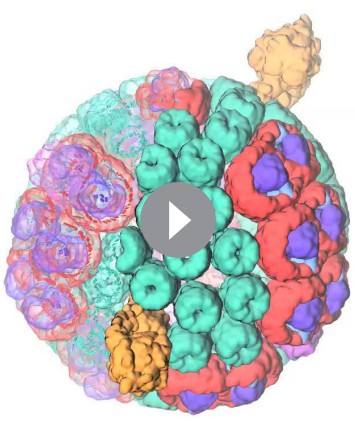

**Video 1.** Chromatophore structural model. A movie that shows a detailed structural model for the low-light adapted chromatophore vesicle as displayed in *Figure 1* (*Cartron et al., 2014*). Presented is a rotating view of the vesicle comprising LH2 complexes (green), dimeric RC-LH1-PufX complexes (red-blue-lime green), dimeric cyt$bc_1$ complexes (magenta), and ATP synthases (orange). For half of the model, proteins are shown in solid surface representation, and for the other half, proteins are shown as transparent surfaces with bacteriochlorophylls (BChls), represented by their porphyrin rings, shown as solid surfaces.

In addition to ATP synthesis, the chromatophore utilizes the generated proton motive force also for NADH production via NADH dehydrogenase (*Klamt et al., 2008*) and, thereby, for control of the quinone/quinol pool redox state. Other channels for proton gradient depletion are flagellar motility (*Kojadinovic et al., 2013*) and proton leak across the vesicle membrane. In the present study, we focus on the overall energy conversion characteristics of the molecular components identified in the current structural model (*Cartron et al., 2014*), namely LH2, RC-LH1, cyt$bc_1$, and ATP synthase, where NADH dehydrogenase plays an indirect role.

Efficient energy conversion requires some degree of robustness with respect to supramolecular organization, since no two chromatophores are likely to be identical. Though chromatophore vesicles share structural motifs (*Bahatyrova et al., 2004*; *Cartron et al., 2014*) that vary gradually with growth conditions, inevitable irregularities in the distribution of their constituent proteins and their quinone/quinol pools render chromatophores heterogeneous, requiring energy conversion processes to be insensitive to structural inhomogeneity. Robustness in photosynthetic systems had been demonstrated computationally for the excitation transfer step of light harvesting at the single protein level with respect to loss or rearrangement of pigments (*Şener et al., 2002*) as well as against fluctuations of pigment site-energies (*Damjanovic et al., 2002*) and at the vesicle level against deformations of the pigment network (*Sener et al., 2010*).

Efficiency of energy conversion in a photosynthetic system is not straightforward to define, since it involves multiple interrelated subprocesses spanning both quantum mechanical and classical domains over timescales ranging from picoseconds to milliseconds (*Blankenship, 2014*; *van Amerongen et al., 2000*). A simple measure of conversion efficiency at the light harvesting stage is provided by the quantum yield, $q$, defined as the probability, upon the absorption of a photon by any pigment of the chromatophore, of charge separation at any RC ready for excitation-induced electron transfer to quinone. The quantum yield is solely a function of pigment network geometry, is independent of incident light intensity, and is found to be close to unity (*Strümpfer et al., 2012*; *Sener et al., 2011*) for initial chlorophyll light absorption; in case of carotenoid light absorption the quantum yield can be lower due to so-called covalent electronic excitation as argued in (*Ritz et al., 2000b*). Since excitation transfer does not constitute a rate-limiting step of photosynthetic energy conversion, the quantum yield is not a major limiting factor for the overall efficiency of the chromatophore. A comprehensive measure of chromatophore efficiency that also permits a limited comparison with photovoltaic systems is the conversion efficiency of captured solar energy to the chemical energy of the final photoproduct, namely ATP.

Earlier studies of photosynthetic membrane systems include percolation theory-based models of quinone diffusion in *Rhodospirillum (Rsp.) photometricum* membranes (*Scheuring et al., 2006*), plastoquinone diffusion in thylakoid membranes (*Kirchhoff et al., 2002*), models of dissipative photoprotective behavior in *Rsp. photometricum* membranes (*Caycedo-Soler et al., 2010*), and stoichiometry-based rate kinetics (*Geyer et al., 2007*, *2010*). In fact, long before any structural information of the light harvesting apparatus of purple bacteria was available, Vredenberg and Duysens (*Vredenberg and Duysens, 1963*) postulated that the total fluorescence yield can be expressed in terms of the ratio of closed and open RCs, after which random-walk models of excitation transfer were developed using a master equation formalism (*Den Hollander et al., 1983*). Prior to the

availability of AFM imaging data (*Bahatyrova et al., 2004*), the supramolecular organization of chromatophores was suggested to feature RCs partially surrounded by LH-complexes facilitating efficient shuttling of quinones (*Joliot et al., 1990*, *1996*; *Jungas et al., 1999*).

The aim of the present study is to determine, based on a supramolecular structural model (*Cartron et al., 2014*), for the chromatophore of *Rba. sphaeroides* the ATP production rate as a function of illumination and vesicle stoichiometry along with the corresponding energy conversion efficiency. A low-light adapted chromatophore vesicle model is considered (*Cartron et al., 2014*), since low-light illumination, namely ≤10% of full sunlight, is typical for the habitat of purple bacteria (*Woronowicz and Niederman, 2010*; *Blankenship, 2014*). The quantum yield of excitation transfer for the pigment network geometry shown in *Figure 1* is determined in terms of an effective Hamiltonian formulation. The processes subsequent to charge separation and the corresponding rate kinetics of ATP production are described in terms of chromatophore vesicle stoichiometry, instead of at atomic detail, by identifying rate-limiting steps. The organizational optimization of the chromatophore is considered in terms of the dependence of energy conversion on vesicle composition and illumination conditions.

## Results

Based on the theoretical framework discussed in the Materials and methods section below, one can quantify how well the chromatophore performs in converting light energy into ATP synthesis and compare its performance characteristics, such as energy conversion efficiency, to the characteristics of other biological and artificial energy conversion systems. In particular, we examine below ATP turnover of the chromatophore as a function of light intensity and vesicle composition. The reader is urged to read the Materials and methods (Section 4) before proceeding further with the present section.

### ATP turnover rate as a function of illumination

Previous studies showed that the quantum yield of excitation transfer, $q$, computed through *Equation 8* below and discussed in greater detail in Supplementary Materials, is very high, namely, 85–94%, varying gradually with LH2:RC stoichiometry (*Şener et al., 2007*, *2010*, *2011*). For the vesicle presented in *Figure 1* the quantum yield, $q$, has a value of 0.91, consistent with earlier studies (*Şener et al., 2007*, *2010*; *Cartron et al., 2014*). Such high value for the quantum yield, close to the ideal limit of 1, is achieved because loss due to internal conversion and fluorescence arises much more slowly (rates about $(1 \text{ ns})^{-1}$) than excitation transfer or charge separation at RC (rates about $(10 \text{ ps})^{-1}$). Clearly, the quantum yield does not constitute a limiting factor for the overall energy conversion efficiency in the chromatophore.

At very low light intensity, nearly all electronic excitation delivered to RCs contribute to the generation of a proton gradient across the membrane and to eventual ATP synthesis. With increasing light intensity, the cycling time of quinones at the RC, $\tau_{RC}(I)$ as given by *Equation 19*, increases; fewer RCs are found in a state available to receive photoexcitation, described by the probability, $p_{RC}(I)$, given by *Equation 13*, and resulting in a corresponding loss of electronic excitation. The time-scale with which the quinone/quinol pool redox state adapts to a change in light conditions is reported to be about 0.5 s (*Woronowicz et al., 2011a*).

The ATP turnover rate, $k_{ATP}$, calculated according to *Equation (20)*, and the energy conversion efficiency, $\eta_{ATP}$, calculated according to *Equation (21)*, for a low-light adapted chromatophore vesicle (*Figure 1*) under steady-state illumination are presented in *Figure 2*. At light intensities equivalent to 1% and 3% of full sunlight, the vesicle is found to produce ATP molecules at a rate of 82 s⁻¹ and 118 s⁻¹, respectively. At the high-light limit, the ATP synthesis rate approaches 158 molecules s⁻¹. These rates are consistent with experimental observations for continuous light-induced photophosphorylation, reported to be in the range of 0.017 molecules per BChl per second (*Saphon et al., 1975*) and 0.05 ATP molecules per BChl per second (*Clark et al., 1983*), corresponding to ~43 ATP molecules s⁻¹ and ~130 ATP molecules s⁻¹, respectively, for the vesicle shown in *Figure 1*. We note that the Clark estimate was reported for *Rhodopseudomonas capsulata*. The corresponding energy conversion efficiency, $\eta_{ATP}$, at the stated low-light intensities of 1% and 3% of full sunlight, calculated in the present study, are 12% and 7%, respectively. Notably, an upper-limit of 30% was estimated in (*Hellingwerf et al., 1993*) for the conversion efficiency of photosynthesis in

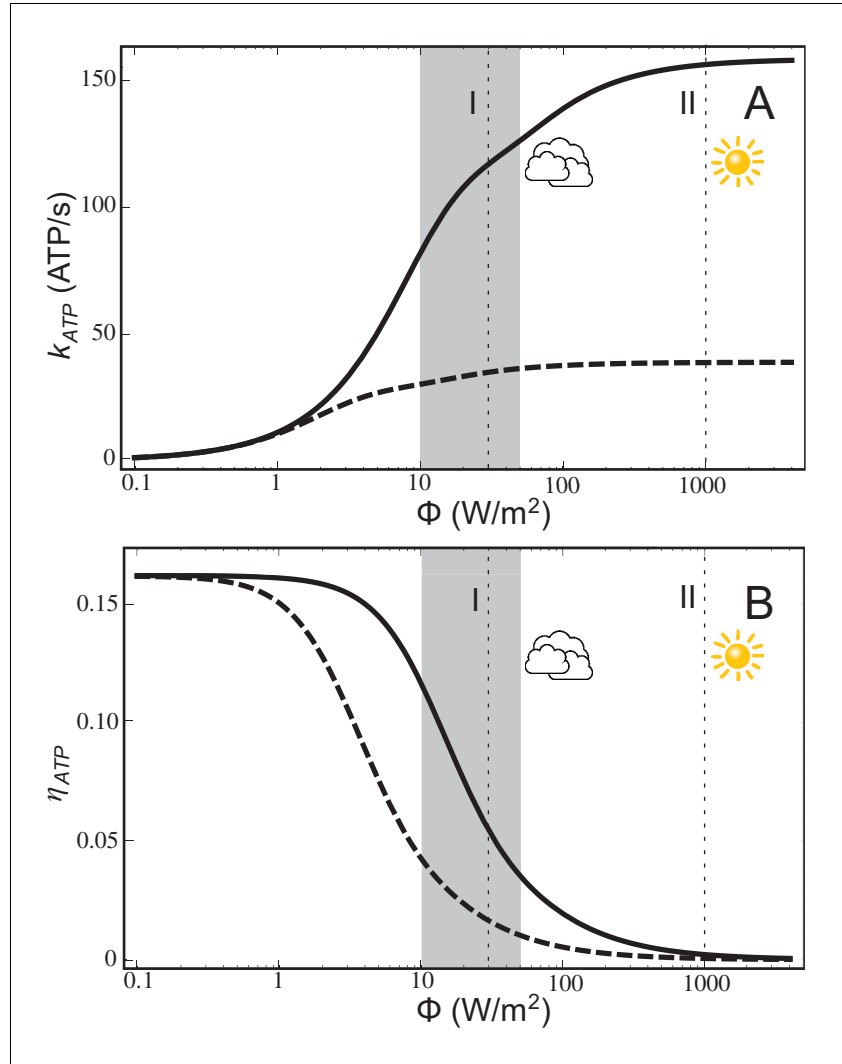

**Figure 2.** ATP production rate and energy conversion efficiency. (**A**) Steady-state ATP production rate, $k_{ATP}$, calculated according to **Equation (20)**, and (**B**) energy conversion efficiency, $\eta_{ATP}$, calculated according to **Equation (21)**, of a chromatophore vesicle as a function of incident light intensity $\Phi$. Solid curves correspond to the vesicle shown in **Figure 1**; dashed curves represent a similar vesicle with only a single cyt$bc_1$. The vertical lines denote the light intensities corresponding to (I) 3% of full sunlight (30 W/m$^2$), a typical growth condition for purple bacteria, and (II) full sunlight (1 kW/m$^2$), respectively. Thus, for light intensities typical for the habitat of purple bacteria (1–5% of full sunlight; shaded area) the energy conversion efficiency $\eta_{ATP}$ of a chromatophore vesicle is between 0.12–0.04.

*Rba. sphaeroides*. In comparison, the efficiency value, $\eta_{ATP}$, computed for the recently established model (**Cartron et al., 2014**) of the chromatophore (**Figure 2**) ranges between 0%–17%.

The lower efficiency values, $\eta_{ATP}$, for the chromatophore at higher light intensities (**Figure 2B**) does not indicate a failing, since the chromatophore does actually produce slightly more ATP in high-light than in low-light illumination (**Figure 2A**), but rather reflects the optimization of purple bacteria for a low-light intensity habitat.

The saturation of ATP synthesis with increasing light intensity seen in **Figure 2** arises because quinol turnover capacity at the cyt$bc_1$ complex becomes rate limiting at higher light intensities. The rate limiting property of cyt$bc_1$ complexes was suggested by earlier studies (**Lavergne et al., 2009**; **Geyer et al., 2010**) and is discussed below in connection with **Equation (14)** in Materials and methods. The maximal electron processing capacity of all cyt$bc_1$ complexes is estimated (see

Equation (14) in Materials and methods) to be $2 \times n_B \tau_B^{-1} = 320 \sim \mathrm{s}^{-1}$, where $n_B = 4$ is the number of cyt$bc_1$ dimers and $\tau_B$ = 25 ms is the quinol turnover time at cyt$bc_1$ (*Crofts, 2004*) and the prefactor 2 accounts for every quinol transferring two electrons. The electron processing capacity at cyt$bc_1$ becomes equal to the total RC electron turnover rate, $Iq$, at a light intensity of 9 W/m$^2$, i.e., at approximately 1% of full sunlight. As illumination exceeds this low-light value, RC electron turnover is limited by the electron processing capacity of cyt$bc_1$, leading to a gradual saturation of proton gradient formation and ATP turnover, as seen in *Figure 2A*, thereby reducing the efficiency of ATP synthesis (*Figure 2B*).

Rate limitation of ATP synthesis by the cyt$bc_1$ turnover capacity can be related also to the availability of quinones at the RC. In the absence of bound quinone, excitations delivered to a RC are wasted, except if the excitation escapes from the RC and reaches another RC ready for quinone reduction, especially within the same RC-LH1 dimer. However, with increasing illumination the likelihood of nearby RCs having quinones available also diminishes, excitation energy is lost, and energy conversion efficiency is reduced. The probability of a RC being ready for quinone reduction, $p_{\mathrm{RC}}(I)$, is given by *Equation (13)* in the Materials and methods section. With increasing illumination, $p_{\mathrm{RC}}(I)$ decreases, thereby reducing the overall conversion efficiency, $\eta_{\mathrm{ATP}}$. At 1% of full sunlight, $p_{\mathrm{RC}}(I)$ assummes the value of 0.73, which, according to *Equation (13)*, drops to 0.23 at 5% of full sunlight. Remarkably, the role of closed and open RCs in determining the overall efficiency of the photosynthesic apparatus had been already pointed out long before any structural details were known (*Vredenberg and Duysens, 1963*).

The rate limiting effect of cyt$bc_1$ can be further illustrated considering the efficiency of chromatophores with fewer cyt$bc_1$ complexes compared to the ones shown in *Figure 1*. As indicated by the dotted lines in *Figure 2A,B*, describing ATP synthesis in a chromatophore with a single cyt$bc_1$ dimer, a lower number of cyt$bc_1$ dimers results in a reduction of the ATP production rate, $k_{\mathrm{ATP}}$, and, accordingly, in a lower conversion efficiency, $\eta_{\mathrm{ATP}}$.

A comparison with plant light harvesting efficiencies is not straightforward: efficiency for biomass production is significantly lower than the aforementioned thermodynamic efficiency; in fact, only as little as 1% of total incident solar energy is stored by crop plants as biomass (*Blankenship et al., 2011*).

One might wonder how the chromatophore compares to engineered photovoltaic devices. At peak solar intensity photovoltaic-driven electrolysis is reported to have an energy conversion efficiency of 5–15% (*Blankenship et al., 2011*). However, comparison of efficiency alone overlooks issues such as stability and reclaimability of the energy stored in the final products. More refined measures of efficiency need to include total integrated cost of components, life expectancy, repair and maintenance.

## Optimality of vesicle composition for ATP production

Evolutionary pressure toward greater fitness at the organism level results in the composition and architecture of photosynthetic systems to display adaptation toward optimal function (*Xiong et al., 2000*; *Şener and Schulten, 2008*; *Blankenship, 2014*). Such adaptation has been reported for the individual protein level; it is not as well understood at the system integration level. For instance, pigment networks of individual light harvesting proteins were reported to display optimality and robustness in their quantum yield with respect to the spatial organization of pigments and the site energy distribution (*Şener et al., 2002*; *Noy et al., 2006*; *Damjanovic et al., 2002*); a similar robustness was reported with respect to size-scaling deformations of an entire vesicle (*Sener et al., 2010*). Prior studies did not take into account optimization of the complete energy conversion process, including ATP synthesis, the effects of vesicle composition influenced by growth conditions such as light intensity (*Niederman, 2013*; *Woronowicz et al., 2013*), the regulation of the redox state of the quinone/quinol pool (*Klamt et al., 2008*), or the effects of cell-scale concentration and connectivity of chromatophores also influenced by light intensity at growth (*Tucker et al., 2010*).

In the following, the effect of vesicle composition on the ATP turnover rate is examined in order to determine the degree of optimality of the vesicle composition for ATP production. The vesicle shown in *Figure 1* is used as a reference point for comparison with chromatophores of alternate composition, As composition variables, the number of dimeric cyt$bc_1$ complexes, $n_{\mathrm{B}}$, and the number of dimeric RC-LH1-PufX complexes, $n_{\mathrm{L}}$, are considered for a two-parameter $(n_{\mathrm{B}}, n_{\mathrm{L}})$ family of vesicles with the same surface area as the reference vesicle (*Figure 1*). The dependence of the steady-state

ATP turnover rate, $k_{\mathrm{ATP}}(n_{\mathrm{B}}, n_{\mathrm{L}}; I)$, on $n_{\mathrm{B}}$ and $n_{\mathrm{L}}$ is determined according to *Equations 17,19,20*, where $n_{\mathrm{L}} = 2 \times n_{\mathrm{RC}}$. In order to avoid massive computation, vesicles are not constructed explicitly. Instead, the corresponding quantum yield $q$ is estimated by a linear interpolation on the LH2:RC stoichiometry based on earlier reported values (*Şener et al., 2007, 2010, 2011*) as described in Materials and methods (*Equation 10*). Since $q$ varies very little with vesicle composition, the dependence of $k_{\mathrm{ATP}}$ on composition is dominated primarily by the explicit $n_{\mathrm{B}}$ and $n_{\mathrm{RC}}$ dependence in *Equations 19,20*.

The rate $k_{\mathrm{ATP}}(n_{\mathrm{B}}, n_{\mathrm{L}}; I)$ is shown in *Figure 3* for light intensities equal to 1% and 3% of full sunlight. The respective ATP synthesis rates for the reference vesicle in *Figure 1* under these two illumination conditions are 82 and 118 ATP molecules/s, respectively, (marked by circles in *Figure 3*) which corresponds to 79% and 50% of the maximum possible rate (marked by crosses in *Figure 3*) among all possible $(n_{\mathrm{B}}, n_{\mathrm{L}})$ values at that illumination. Clearly, steady state ATP synthesis is not optimized by the vesicle composition shown in *Figure 1*. The turnover rate, $k_{\mathrm{ATP}}$, would be improved by an $n_{\mathrm{B}} : n_{\mathrm{L}}$ ratio that is greater than the native value of $1 : 3$ (*Crofts, 2004*; *Cartron et al., 2014*), as suggested also by a comparison of the turnover times at cyt$bc_1$ and RC ($\tau_B/\tau_L \simeq 8$).

A reason for the aforementioned suboptimal $(n_{\mathrm{B}}, n_{\mathrm{L}})$ values in native low-light adapted vesicles might be protection against light-induced damage that can arise at high illumination via destruction of the vesicle membrane through overacidification. Though typical illumination levels in habitats of purple bacteria are low, occasional surges in light intensity are inevitable. During sustained (>1 s) high illumination intervals, a proton turnover unhindered by a low ($n_{\mathrm{B}} = 4$) cyt$bc_1$ stoichiometry can exceed the turnover capacity of the ATP synthases, resulting in overacidification of the vesicle interior, harming the integrity of the chromatophore membrane and its proteins. The observed $n_{\mathrm{B}}$ value of 4, apparently suboptimal for most light intensities (*Figure 3*), ensures that during sustained overillumination proton turnover is limited by cyt$bc_1$ to a rate below the synthesis capacity of ATP synthase (*Lavergne et al., 2009*; *Geyer et al., 2010*), thus preventing overacidification.

The $(n_{\mathrm{B}}, n_{\mathrm{L}})$ value also has an effect on the size of the quinone/quinol pool relevant for intermittent energy storage under fluctuating light conditions, since the number of quinones in the system correlates with the number of RCs (*Comayras et al., 2005*; *Woronowicz et al., 2011a*; *Cartron et al., 2014*). Energy conversion through the quinone/quinol pool also involves electron exchange processes from outside the chromatophore as furnished, for example, through the enzymes NADH dehydrogenase and succinate dehydrogenase (*Klamt et al., 2008*).

The turnover capacities of cyt$bc_1$ and ATP synthase are compared in Materials and methods in relation to the rate limitation of energy conversion by the cyt$bc_1$. A single ATP synthase is sufficient to take advantage of proton turnover of an entire chromatophore (*Etzold et al., 1997*). Additional ATP synthases reported in chromatophore vesicles (*Cartron et al., 2014*) appear to provide necessary redundancy, since an isolated chromatophore without ATP synthase is non-functional. In this regard, it is of interest that vesicles have been found to occasionally fuse through formation of membrane tubes (*Tucker et al., 2010*) permitting passage of protons between neighboring chromatophore vesicles, thereby sharing their proton gradients with the ATP synthases of multiple vesicles, reducing the need for back-up ATP synthases and even permitting less than one ATP synthase per vesicle.

Robustness requirements for protecting the vesicle against damage under environmental strain apparently supersede optimality constraints for steady state conditions. A photosynthetic vesicle adapted for steady-state illumination at higher light intensities than considered in this study would require a larger number of cyt$bc_1$ to maximize ATP production, along with more than the 1–2 ATP synthases observed per vesicle (*Cartron et al., 2014*).

## Discussion

The combined structural and functional model of a low-light adapted chromatophore (*Cartron et al., 2014*) permits a quantitative description of ATP synthesis at different light intensities. The energy conversion efficiency, $\eta_{\mathrm{ATP}}$, is determined to be ~12%–4% at the low-light conditions typical for purple bacterial habitats (1%–5% of full sunlight), dropping rapidly to ≤0.1% beyond full sunlight conditions. Moderate levels of illumination saturate the bacterial light harvesting apparatus lowering its efficiency, whereas plants and photovoltaic devices function efficiently at high

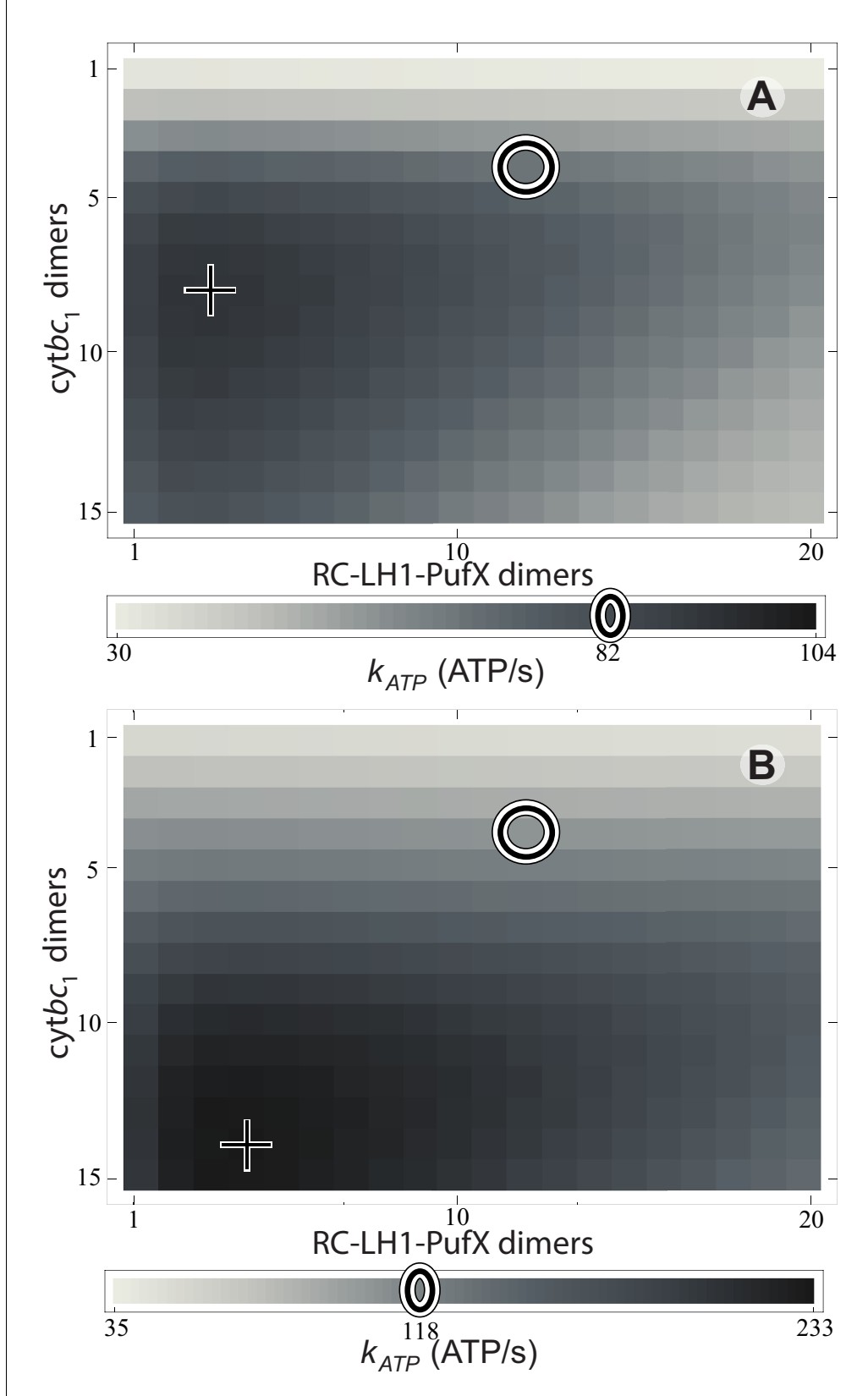

**Figure 3.** Effect of vesicle composition on steady-state ATP production at different light intensities. Vesicle composition is given in terms of the number of cyt$bc_1$ dimers ($n_B$) and of RC-LH1-PufX dimers ($n_L$) for vesicles featuring identical surface area; LH2 composition of the vesicle is determined by

*Figure 3 continued*

considering the vesicle shown in *Figure 1* as a reference point and adjusting the number of LH2 complexes to compensate for the changes in the number of cyt$bc_1$ and RC-LH1-PufX dimers to cover the vesicle surface. ATP production rate, $k_{ATP}$, is shown for (**A**) 1% of full sunlight (10 W/m$^2$) and (**B**) 3% of full sunlight (30 W/m$^2$), determined according to *Equation (20)*. The two RC-LH1-PufX monomers of the vesicle in *Figure 1* were counted as a single dimer for the purposes of this plot. The reference vesicle (*Figure 1*) is represented by a circle, corresponding to an ATP production rate of 82 s$^{-1}$ (118 s$^{-1}$), i.e. 79% (51%) of the maximum possible rate among all stoichiometries, for 1% (3%) of full sun light. The optimal vesicle composition for each illumination is represented by a cross; the corresponding LH2 count for optimal composition is 93 (74) at 1% (3%) of full sunlight as compared with 63 for the reference vesicle (circle). The ATP production rate is marginally greater for vesicles that contain more cyt$bc_1$ and LH2 complexes at the expense of fewer RC-LH1-PufX complexes as compared with the reference vesicle. This increase in ATP production rate results from cyt$bc_1$ being the rate-limiting component in the energy conversion process.

light intensities. The efficiency curve determined in the present study for the purple bacterial chromatophore (*Figure 2B*) indicates specialization for low-light intensities.

The primary rate-limiting component among the energy conversion steps in the chromatophore appears to be quinol turnover at cyt$bc_1$, as discussed in Section 4.2. The rate limitation at cyt$bc_1$, as compared with the ATP synthase turnover capacity, prevents the generation of an overly strong proton gradient at sustained high-light conditions, thereby protecting the chromatophore against over-acidification of its interior and assuring vesicle integrity. As the light intensity $I$ increases, photoexcitations are more likely to be dissipated as the probability for a RC to have a quinone or semiquinone ready to accept an electron, $p_{RC}(I)$, decreases.

The chromatophore composition appears to be suboptimal for ATP production under steady-state illumination. The chromatophore is apparently a highly specialized device that performs its energy conversion function robustly for low average light intensity, while featuring protective measures that dissipate energy at higher light intensity. Robustness against damage, such as overacidification of the membrane due to sustained overillumination, appears to supersede optimality under idealized conditions, such as steady state illumination.

The present study focuses on steady state energy harvesting in the chromatophore without explicitly modeling the spatial dynamics of the charge carriers (quinone/quinol and cytochrome $c_2$), the redox states of the proteins (RC and cyt$bc_1$), proton leakage through the membrane, or the coupling of NADH dehydrogenase to the proton-motive force. A more complete description of chromatophore function requires placement of NADH dehydrogenase, along with possibly succinate dehydrogenase, cytochrome $c$ oxidase, and ubiquinol oxidase in the chromatophore membrane, the presence of which would also affect the energy conversion efficiency determined in this study. The added enzymes need to be described along with their reactions with redox partners located in the cell's cytoplasm. In particular, a non-steady state formulation is necessary to account for spatial heterogeneity and light intensity dependence of the redox states of the proteins and the charge carriers in the chromatophore.

The present study differs from earlier studies in functional modeling of the chromatophore (*Geyer et al., 2007*, *2010*) in several respects: first, it is based on an explicit atomic-detail structural model; second, instead of employing many (over 30) adjustable parameters, few experimentally determined rate constants are employed to describe the rate determining steps; third, a steady-state description is chosen such that energy conversion steps that are not rate-limiting can be left out of the kinetic model. Nonetheless, earlier and present studies give similar results for the overall ATP synthesis rate at saturation, since this rate is determined largely by the total turnover capacity of cyt$bc_1$ complexes as a rate-limiting component.

A key role in chromatophore energy conversion involves proton translocation, generating and using proton motive force. The present treatment does not resolve individual translocation steps, but rather assumes that the individual steps taking place at the overall RC, cyt$bc_1$, and ATP synthase proton reactions can be treated as a single reaction event. Primary conclusions reached presently would not be affected by a more detailed description, i.e., the model is robust with respect to the neglect of explicit modeling of individual proton translocation steps and proton motive force conversion.

Integrative models of organelle function such as the one presented here provide a bridge between experimental methods that do not resolve temporal and spatial detail needed for

establishing physical mechanisms and microscopic simulations that span the multiple length and time scales relevant for the function of living cells.

## Materials and methods

In the following, structural organization and energy conversion in the chromatophore are described in terms of a kinetic model. It is highly recommended that the text below is read before Sections 2 and 3. First, the supramolecular organization of a low light-adapted chromatophore vesicle is introduced. Next, the energy conversion processes are characterized: excitation transfer, diffusion of quinones/quinol and of cytochrome $c_2$, and ATP synthesis. The description is based on steady state kinetics. Inhomogeneities of the quinone/quinol and cytochrome $c_2$ pools and of the membrane proton gradient are not modeled; instead, the three attributes are assumed to function as homogeneous buffers of energy storage. The framework outlined is used to define three different measures of efficiency for the chromatophore: (i) quantum yield, $q$, (ii) quinol conversion ($Q \rightarrow QH_2$) probability, $\eta_Q$, i.e., the probability that an absorbed photon is successfully utilized for quinol formation at a RC, and (iii) energy conversion efficiency, $\eta_{ATP}$, i.e., the ratio of the energy stored in the conversion ADP→ATP to incident solar energy absorbed.

### Supramolecular organization of a chromatophore vesicle adapted to low-light illumination

As already stated, the structural model of the chromatophore considered in the present study is a variation of the model reported in (*Cartron et al., 2014*). The primary components of chromatophore vesicles in purple bacteria, as depicted in *Figure 1*, are, in order of energy utillization (*Cogdell et al., 2006*,*Cartron et al., 2014*): (i) light harvesting complex 2 (LH2) (*Koepke et al., 1996*; *Papiz et al., 2003*); (ii) light harvesting complex 1 (LH1 [*Qian et al., 2008*; *Sener et al., 2009*]); (iii) RC (*Jamieson et al., 2002*; *Strümpfer and Schulten, 2012a*); (iv) cyt$bc_1$(*Crofts, 2004*; *Crofts et al., 2006*); and (v) ATP synthase (*Feniouk and Junge, 2009*; *Hakobyan et al., 2012*). RC-LH1 complexes typically form dimeric RC-LH1-PufX complexes facilitated by the polypeptide PufX (*Qian et al., 2013*; *Sener et al., 2009*), although monomeric complexes are also found in membranes from photosynthetically grown cells at a ratio of approximately 10% (*Olsen et al., 2008*). The chromatophore in *Figure 1* exhibits for the LH2:RC complexes a stoichiometry of 2.6:1 and corresponds to a low-light-adapted vesicle as described in (*Sener et al., 2010*; *Cartron et al., 2014*). In a typical vesicle, about a hundred protein complexes, LH2 and RC-LH1-PufX, form an efficient light harvesting network (*Şener et al., 2007*, *2010*) supplying electronic excitation energy for the conversion of quinones to quinols. The quinols produced at the RC are converted back to quinones by cyt$bc_1$ to generate a proton gradient across the chromatophore vesicle membrane, which, in turn, is consumed by the ATP synthase for the synthesis of ATP from ADP and phosphate. The electrons from quinol-to-quinone conversion are shuttled back to the RC by cytochrome $c_2$ acting inside the vesicle. These energy conversion processes are illustrated in *Figure 4*. We note that the experimental data (*Saphon et al., 1975*; *Clark et al., 1983*) used to test the present energy conversion model based on (*Cartron et al., 2014*) were not obtained with chromatophores in vivo, but for a suspension of chromatophores in a pH-buffer; the energy conversion processes as coupled to the entire bacterium are inevitably more complex than portrayed here.

Atomic level structural models of chromatophores have been presented earlier (*Şener et al., 2007*, *2010*; *Hsin et al., 2010b*; *Sener et al., 2011*; *Chandler et al., 2014*) for *Rba. sphaeroides* and *Rsp. photometricum* and their mutants. The supramolecular organization of the vesicles in *Rba. sphaeroides* was determined primarily by AFM and EM images of intact membrane domains (*Bahatyrova et al., 2004*; *Frese et al., 2004*; *Scheuring et al., 2007*; *Olsen et al., 2008*; *Qian et al., 2008*; *Scheuring and Sturgis, 2009*), whereas the stoichiometry of light harvesting proteins was determined by optical spectroscopy (*Sener et al., 2010*) and mass spectrometry (*Cartron et al., 2014*). Vesicle models were subsequently constructed by mapping planar membrane patches viewed through AFM imaging back onto the parent spherical domains (*Şener et al., 2007*), adjusting for the observed packing density (*Olsen et al., 2008*) and the spatial arrangement patterns (*Hsin et al., 2009*; *Qian et al., 2008*) of the constituting proteins.

The chromatophore model shown in *Figure 1* comprises, in addition to the aforementioned constituent proteins, 16,000 lipids and 900 quinones, corresponding to a system containing 100 million

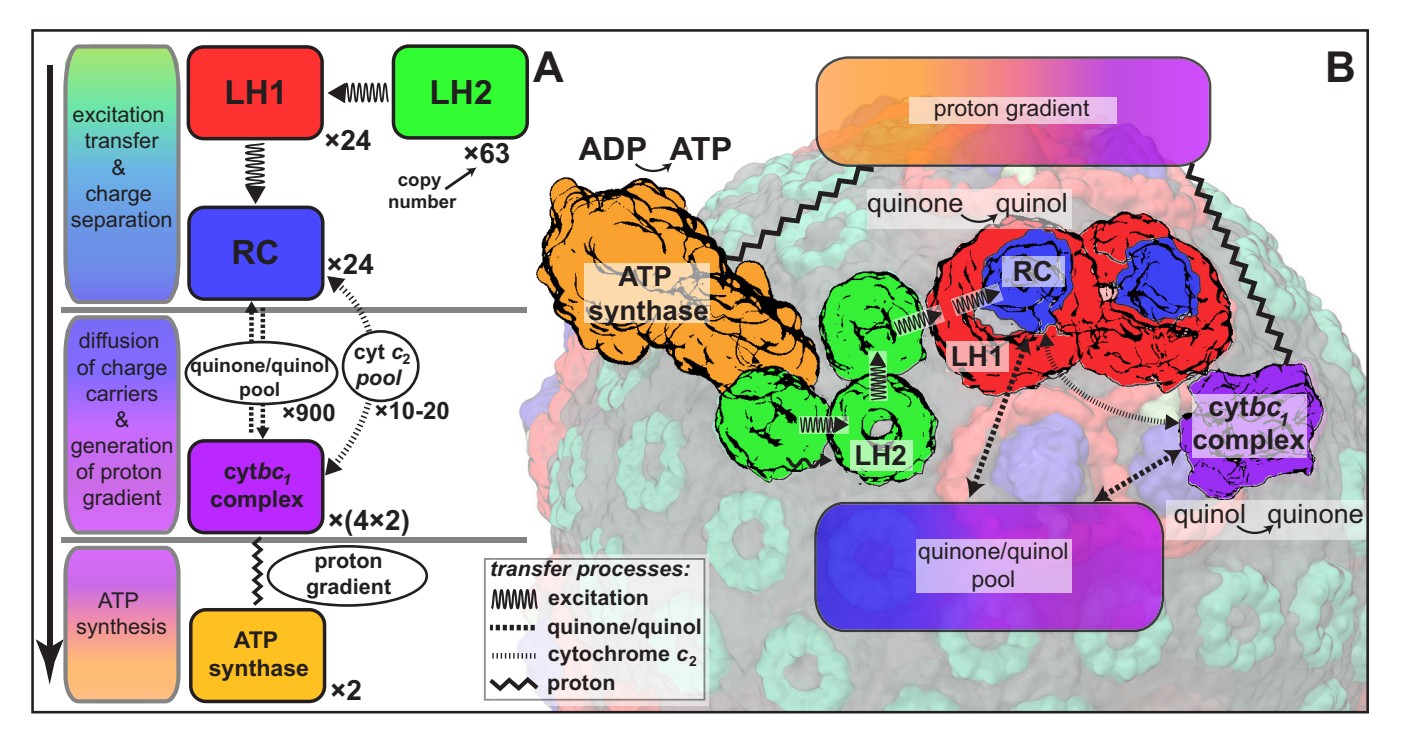

**Figure 4.** Processes involved in energy conversion in the photosynthetic chromatophore. (**A**) Energy conversion processes starting after initial light absorption are divided into three stages: (1) quinol production at RC as a result of excitation transfer; (2) diffusion between RC and cyt$bc_1$ of quinone/quinol and cytochrome $c_2$, together with quinol-to-quinone conversion resulting in a proton gradient across the vesicle membrane; (3) utilization of proton gradient for ATP synthesis. (**B**) Chromatophore components, in which stages (1–3) take place, include LH2 (green), LH1(red)-RC(blue), cyt$bc_1$ (purple), and ATP synthase (brown) complexes as well as the lipid phase (olive; see also *Figure 1B*).

atoms, including solvent. This system has been equilibrated through a 100 ns MD simulation to test the viability of the model employed. However, molecular dynamics simulations of the chromatophore describing energy conversion processes are not considered in the current study, because the large system size combined with timescales of energy conversion reaching milliseconds render a straightforward simulation prohibitive. Instead, the current study aims to describe key rate limiting components of energy conversion processes, such as quinone diffusion and turnover at cyt$bc_1$ as discussed below, to guide future simulation efforts. The atomic detail model is used below for the computation of the quantum yield, but rate kinetics subsequent to charge separation is described in terms vesicle stoichiometry only, with key rate constants taken from experimental studies.

Early chromatophore models prior to (*Cartron et al., 2014*) account only for LH proteins, whereas in proteomics studies, hundreds of different types of non-LH peptides are actually identified, including ATP synthase, cyt$bc_1$, membrane assembly factors, as well as proteins of unknown function (*Jackson et al., 2012*; *Woronowicz and Niederman, 2010*). Most of these components are notably unresolved in AFM images. Assignment of cyt$bc_1$ was recently achieved through EM and AFM studies using gold nanoparticle labeling, revealing separated regions containing one or more cyt$bc_1$, suggested to be located within lipid- and quinone-enriched membrane domains (*Cartron et al., 2014*). It is plausible that cyt$bc_1$ induces different curvature profiles in membrane domains compared to the LH-rich constant-curvature regions predominant in AFM images. Such a curvature-induced separation of protein domains is also supported by experimental (*Frese et al., 2004*; *Sturgis and Niederman, 1996*) and computational (*Frese et al., 2008*; *Chandler et al., 2009*; *Hsin et al., 2009*, *2010a*) studies that established the role of LH2 and RC-LH1-PufX domains in determining membrane shape. Induced curvature profiles are known to exert a segregating force between different types of proteins in the membrane (*Frese et al., 2008*). Mass spectrometry showed that the RC:cyt$bc_1$

stoichiometry is 3:1 (*Cartron et al., 2014*), consistent with earlier observations (*Crofts, 2004*; *Crofts et al., 2006*), corresponding to approximately 4 cyt$bc_1$ dimeric complexes per vesicle.

Chromatophore vesicles typically contain 1–2 ATP synthases (*Feniouk et al., 2002*; *Cartron et al., 2014*). Proteomics studies suggest preferential co-location of ATP synthase with LH2 subunits (*Woronowicz and Niederman, 2010*). Consequently, ATP synthase locations were assigned to LH2-rich regions of the membrane (*Cartron et al., 2014*).

The low-light adapted vesicle studied here contains 63 LH2 complexes, 11 dimeric and 2 monomeric RC-LH1-PufX complexes, 4 dimeric cyt$bc_1$ complexes, and 2 ATP synthases, in a spherical vesicle of 50 nm inner diameter based on a variation of the model reported in (*Cartron et al., 2014*) and shown in *Figure 1* (see also *Video 1*).

Transmembrane proteins beyond those of the light harvesting-cyt$bc_1$-ATP synthase model shown in *Figure 4*, namely NADH dehydrogenase, succinate dehydrogenase, cytochrome $c$ oxidase, and ubiquinol oxidase, are associated with controlling the redox state of the quinone/quinol pool in the chromatophore (*Klamt et al., 2008*). These proteins, presented schematically in *Figure 5*, indirectly couple the chromatophore proton gradient to metabolic reactions in the cytoplasmic part of the bacterial cell. Indeed, the chromatophore structure shown in *Figure 1* may accommodate, by removal of LH2 complexes near cyt$bc_1$ complexes (the latter referred to as complex III in the respirasome of mitochondria [*Dudkina et al., 2011*]), the placement of adjacent NADH dehydrogenase complexes (referred to as complex I) in an arrangement similar to that in respirasomes as reported in (*Dudkina et al., 2011*). First simulations, employing the complex I structure reported in (*Baradaran et al., 2013*), have demonstrated that the chromatophore can adapt to the necessary local shape change.

## Description of energy conversion in the chromatophore

The overall aim of the present study is to determine the ATP synthesis rate as a function of chromatophore vesicle illumination and composition establishing, thereby, the energy conversion efficiency. The three stages of energy conversion in the chromatophore introduced above are summarized in *Figure 4*. These stages span time scales ranging from femto- and picoseconds

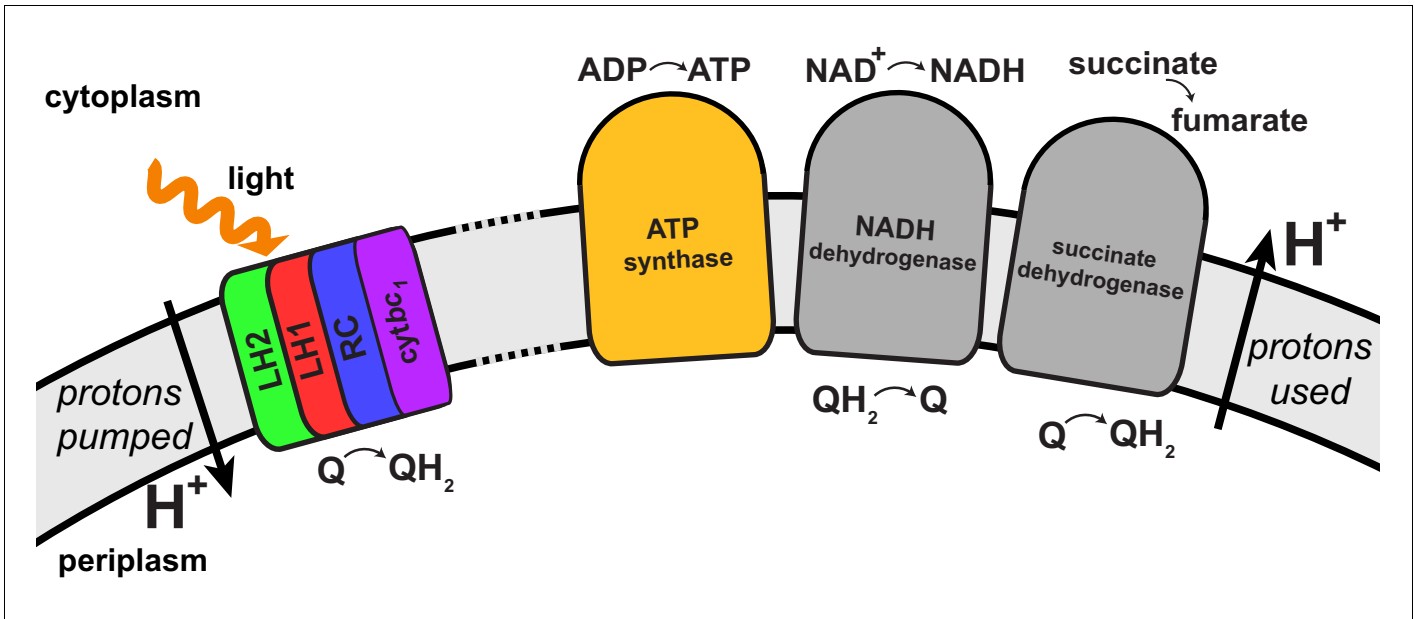

**Figure 5.** Regulation of the quinone/quinol-pool redox state in the chromatophore involves transmembrane proteins beyond those included in the light harvesting-RC-cyt$bc_1$-ATP synthase system described in (*Cartron et al., 2014*) and shown in *Figure 4*. Succinate dehydrogenase and NADH dehydrogenase regulate the quinone/quinol redox state in the chromatophore membrane. This regulation influences the light intensity dependence of the ATP production rate in the chromatophore by changing the likelihood of finding available quinones at the RC. Two further proteins involved in redox kinetics are cytochrome $c$ oxidase and ubiquinol oxidase, which are not shown.

(transfer of excitations) to milliseconds (diffusion of quinols, quinones, and cytochrome $c_2$; ATP synthesis), involving both classical and quantum dynamics.

The absorbed light power $I$ is given in units of photons absorbed per second for the entire vesicle, i.e., it holds approximately, $I = F\sigma_{total}$ where $F$ is the flux of useable photons and $\sigma_{total}$ is the total absorption cross-section of the chromatophore determined via the functional absorption cross-section reported in (**Woronowicz et al., 2011a**).

Key quantities describing energy conversion are the quinone-to-quinol formation rate $k_{Q\rightarrow QH_2}(I)$, the quinol-to-quinone use rate $k_{QH_2\rightarrow Q}(I)$ and the ATP synthesis rate, $k_{ATP}(I)$, all functions of the absorbed light power $I$. For stationary illumination, assumed here, the chromatophore kinetics becomes stationary and, as a result, the rates $k_{Q\rightarrow QH_2}(I)$, $k_{QH_2\rightarrow Q}(I)$, and $k_{ATP}(I)$ must be identical,

$$k_{Q\rightarrow QH_2}(I) = k_{QH_2\rightarrow Q}(I) = k_{ATP}(I). \tag{1}$$

We note here that every net quinol→quinone conversion event at cyt$bc_1$, due to the so-called Q-cycle (**Crofts, 2004**), results in the release of four protons into the vesicle interior (two at cyt$bc_1$ and two at RC, for each quinol passage), which coincidentally happens to be, in the present system, the same number of protons as the ones that have to move back over the membrane to produce one ATP molecule at ATP synthase (based on the assuption of a 12-subunit c-ring of the ATP synthase). Under steady state conditions, the rate, $k_{ATP}(I)$, is then equal to $k_{QH_2\rightarrow Q}(I)$ and $k_{Q\rightarrow QH_2}(I)$.

The quinone/quinol pool in the lipid phase of the vesicle and the proton gradient across the vesicle membrane act as temporary energy buffers between light harvesting and ATP synthesis (**Feniouk and Junge, 2009**; **Clark et al., 1983**). Under the steady-state conditions assumed here, quinone-quinol pool and redox states of RC/cyt$bc_1$ are assumed to feature spatially homogeneous distributions. As a result, individual diffusive processes of quinone/quinol, cytochrome $c_2$, and protons do not need to be modeled, and the aforementioned energy buffers are determined solely by incident light intensity and quinol→quinone turnover capacity of cyt$bc_1$, the latter constituting the rate limiting conversion process as discussed below. Typical proton diffusion timescales are on the order of microseconds (**Agmon, 1995**), i.e., not rate limiting compared to turnover at cyt$bc_1$ and, therefore, do not affect significantly overall conversion rates.

In going beyond steady-state conditions, a simulation of quinone mobility in a lamellar chromatophore membrane has recently been achieved for a 20 million atom, 150 ns simulation (**Chandler et al., 2014**); however, the time scale covered is not long enough to observe long-range positional relaxation of the quinone/quinol pool. This non-stationary behavior needs to be addressed by a coarser description like the ones employed for cell-scale modeling (**Roberts et al., 2013**).

In purple bacteria there are proton gradient consumption channels other than ATP synthesis. These channels include: flagellar motility (**Kojadinovic et al., 2013**); NADH/NADPH synthesis (**Klamt et al., 2008**; **Blankenship, 2014**) involving respiratory protein complexes in the chromatophore vesicle; leakage across the membrane. These channels are not included in the chromatophore kinetics described below, though the influence of NADH dehydrogenase is implicitly accounted for as explained below.

## Stage I: Light absorption, excitation energy transfer, and quinol formation

The first stage of energy conversion in the chromatophore begins with light absorption by carotenoid and BChl pigments in the light harvesting complexes LH1 and LH2 leading to electronic excitation of individual pigments. Carotenoids transfer excitation within less than a picosecond to a nearby BChl (**Damjanović et al., 1999**; **Berera et al., 2009**) and also play a role in quenching triplet states of BChls through reverse excitation transfer (**Ritz et al., 2000a**). The electronic excitations of BChls embedded in LH1 and LH2 are reviewed in (**Hu et al., 1998**, **2002**; **Cogdell et al., 2006**; **van Grondelle and Novoderezhkin, 2006b**; **Kosztin and Schulten, 2014**). These excitations form so-called exciton states, excitations shared among LH1 or LH2 BChls (**Ma et al., 1997**; **Bradforth et al., 1995**) coherently (**Strümpfer et al., 2012**; **Ishizaki and Fleming, 2009b**; **Rebentrost et al., 2009**). Electronic excitation is transferred efficiently in the form of excitons between light harvesting complexes (**Hu et al., 1997**; **Ritz et al., 1998**; **Janusonis et al., 2008**; **Ishizaki and Fleming, 2009b**).

Exciton-based excitation transfer in the chromatophore proceeds within 10–100 ps (*Sener et al., 2011*), first among the LH2s, then from LH2 to LH1, and finally from LH1 to the four BChls of the RC (*Visscher et al., 1989*; *Beekman et al., 1994*; *Strümpfer and Schulten, 2012a*; *Sener et al., 2009*). In the RC, the excitation quickly settles onto the so-called special pair BChls (*Small, 1995*; *Damjanović et al., 2000*), where it induces the transfer of an electron (*Pawlowicz et al., 2008*; *Jordanides et al., 2004*). This transfer proceeds stepwise to reach a quinone molecule, $Q$, attracted into the RC from the quinone/quinol pool of about 900 molecules (*Cartron et al., 2014*). The quinol and quinone molecules of the pool are inter-converted at RC and cyt$bc_1$ (*Crofts, 2004*) (see *Figure 4*). The electron transferred in the RC is joined on the quinone by a proton, turning $Q$ into semi-quinone, $QH$. Repeating the reaction turns $QH$ into quinol, $QH_2$. In converting $Q$ to $QH_2$ two electron charges move from near the inside of the chromatophore (where the special pair BChls are located and the electron potential is low) to near the cytoplasmic exterior of the chromatophore (where the quinone is bound and the electron potential is high), i.e., to the cytoplasmic side; the protons are attracted from the exterior of the chromatophore vesicle. Freshly formed quinol is released by the RC into the lipid phase of the chromatophore rejoining the quinone/quinol pool.

The efficiency of the 10–100 ps light harvesting step is measured by the so-called quantum yield, $q$, namely the probability that light absorption leads to electron transfer in a RC with a $Q$ or $QH$ bound to receive the electron. The quantum yield can be calculated as reported in (*Sener et al., 2011*, *2010*). Electronic excitation energy absorbed directly or indirectly (through carotenoids) by a BChl is rapidly shared between BChls within individual LH1 and LH2 light harvesting complexes (*Cory et al., 1998*), forming, within about a ps, thermally equilibrated exciton states as established experimentally (*Visser et al., 1996*; *Jimenez et al., 1997*; *Valkunas et al., 2007*) as well as computationally (*Strümpfer and Schulten, 2009*; *Strümpfer et al., 2012*).

The exciton states of the BChls of each complex are determined as eigenstates of the effective Hamiltonian $H^I$, accounting for the $Q_y$ excitations and their coupling inside LH1, LH2 and RC as described in (*Strümpfer et al., 2012*),

$$H^I = \sum_{i=1}^{N_I} E_i^I \,|\, i \,\rangle \langle \,i\, | \,+ \sum_{i>j>0}^{N_1} V_{ij}^I (\,|\, i \,\rangle \langle \,j\, | \,+\, |\, j \,\rangle \langle \,i\, |\,). \tag{2}$$

Here, the index $I$ is employed to label one of the pigment-protein complexes, namely one of 63 LH2s, 24 LH1s, and 24 RC complexes for the vesicle shown in *Figure 1*, with $N_I$ BChls; $|i\rangle$ corresponds to the $Q_y$ excitation of BChl $i$ with excited state energy $E_i^I$; $V_{ij}^I$ accounts for the respective $Q_y$-$Q_y$ coupling among BChls $i$ and $j$. Tables S2 and S3 in Supplementary Materials list the BChl coordinates as well as the constants employed in this study and discuss the computation of the quantum yield in greater detail.

The coupling $V_{ij}^I$ in *Equation (2)* can be computed for well separated pigments ($r_{ij}>1$ nm) using the point-dipole approximation (*Ritz et al., 2001*; *Sener et al., 2011*), employing,

$$V_{ij} = C \left( \frac{\hat{\mathbf{d}}_i \cdot \hat{\mathbf{d}}_j}{r_{ij}^3} - 3 \frac{\left(\hat{\mathbf{d}}_i \cdot \mathbf{r}_{ij}\right)\left(\hat{\mathbf{d}}_j \cdot \mathbf{r}_{ij}\right)}{r_{ij}^5} \right), \tag{3}$$

where $\hat{\mathbf{d}}_i$ is the transition dipole moment unit vector of pigment $i$, $\mathbf{r}_{ij}$ is the vector joining pigments $i$ and $j$; the coupling constant $C$ has the value $C = 348,000\,\text{Å}^3\,\text{cm}^{-1}$ (using wavenumbers as unit of energy) (*Şener et al., 2007*, *2010*). Couplings between closely spaced pigments ($r_{ij}<1$ nm) require quantum chemical calculations as described in (*Damjanović et al., 1999*; *Tretiak et al., 2000*).

The exciton states $|\alpha\rangle = \sum_i c_{i\alpha}|i\rangle$ and the associated energies $\epsilon_\alpha$ correspond to the eigenstates defined through $H^I|\alpha\rangle = \epsilon_\alpha|\alpha\rangle$. As electronic excitations settle within about 1 ps into the Boltzmann-populated excitons (*Strümpfer et al., 2012*; *Strümpfer and Schulten, 2009*), excitation transfer among LH2 and LH1 involves the excitons, not individual chlorophyll or carotenoid excitations. The rate of excitation transfer between a donor complex $I$ and an acceptor complex $J$ is given by (*Ritz et al., 2001*; *Şener et al., 2007*, *2011*)

$$k_{IJ} = \frac{2\pi}{\hbar} \sum_{\mu \in I} \sum_{\nu \in J} p_{\mu}^{I} |(\mu|H_{IJ}|\nu)|^2 J_{\mu\nu}, \tag{4}$$

where $H_{IJ}$ is the matrix of interactions between the excited states of pigments in complexes $I$ and $J$, and

$$J_{\mu\nu} = \int dE\, S_{\mu}^{D}(E) S_{\nu}^{A}(E), \tag{5}$$

is the spectral overlap between donor exciton state $|\mu\rangle$ and acceptor exciton state $|\nu\rangle$ in units of (1/energy) (*Sener et al., 2011*); $S_{\mu}^{I}(E)$ and $S_{\nu}^{I}(E)$ are the normalized ($\int dE\, S(E) = 1$) spectra for emission of the donor ($D$) and absorption of the acceptor ($A$), respectively; $p_{\mu}^{I}$ in *Equation (S6)* are the populations of donor exciton states, which, as stated, become very rapidly (~1 ps) (*Strümpfer and Schulten, 2009*) Boltzmann-distributed such that $p_{\mu}^{I}$ are given by

$$p_{\mu}^{I} = \frac{e^{-\beta\epsilon_{\mu}}}{\sum_{\gamma \in I} e^{-\beta\epsilon_{\gamma}}}. \tag{6}$$

The above description is known as the generalized Förster theory (*Förster, 1948*; *Novoderezhkin and Razjivin, 1996*; *Hu et al., 1997*; *Sumi, 1999*; *Scholes et al., 2001*). For reviews see (*van Grondelle and Novoderezhkin, 2006a*; *Sener et al., 2011*; *Strümpfer et al., 2012*). Excitation transfer kinetics in the chromatophore was reported experimentally in (*Woodbury and Parson, 1984*; *Visscher et al., 1989*; *Crielaard et al., 1994*; *Hess et al., 1994, 1995*).

Exciton migration across the network of light harvesting complexes in the chromatophore can be described by a rate matrix $\mathcal{K}$ which is constructed from inter-complex exciton transfer rates $k_{IJ}$, the latter given by *Equation (S6)*, as follows (*Sener et al., 2010, 2007*)

$$(\mathcal{K})_{IJ} = k_{JI} - \delta_{IJ}\left(\sum_{M} k_{IM} + k_{\text{diss}} + k_{\text{CS}}\,\delta_{I,\text{RC}}\right), \tag{7}$$

where $I,J$ are defined as in *Equation (S6)*; $k_{\text{diss}} = 1/\text{ns}$ is the rate of excitation loss due to internal conversion; $k_{\text{CS}} = 1/(3\,\text{ps})$ is the rate of charge separation at the RC (*Ritz et al., 2001*); $\delta_{I,\text{RC}}$ assumes the value 1 if complex $I$ is a RC and the value 0 otherwise.

The quantum yield, $q$, is the probability for an absorbed photon to initiate charge transfer at a RC ready for electron transfer; $q$ is given for an initial state vector $\mathbf{P}(0)$ by (*Sener et al., 2011, 2007*; *Ritz et al., 2001*)

$$q = -k_{\text{CS}}\,(\mathbf{1}_{\mathbf{RC}})^{T} \cdot \mathcal{K}^{-1} \cdot \mathbf{P}(0) \tag{8}$$

where the components of the vector $(\mathbf{1}_{\mathbf{RC}})$ are $(\mathbf{1}_{\mathbf{RC}})_{I} = \delta_{I,\text{RC}}$; the initial state, $\mathbf{P}(0)$, corresponds to every BChl in the system being equally likely to be excited by photon absorption and accordingly is given by

$$(\mathbf{P}(0))_{I} = N_{I}/(\sum_{J} N_{J}), \tag{9}$$

where $N_{I}$ is the number of BChls in complex $I$ as indicated above. The effect of the initial state, $\mathbf{P}(0)$, on the quantum yield, $q$, arising, for example, due to wavelength-dependent absorption, is considered in (*Şener et al., 2007*), with the result that $q$ is altered by less than 3%. Therefore, wavelength dependence of $q$, through corresponding changes in $\mathbf{P}(0)$, is not considered further in the present study. The quantum yield given by *Equation (8)* for the vesicle shown in *Figure 1* is 0.91.

For alternate vesicle compositions considered in *Figure 3*, the quantum yield $q$ is not computed by an explicit construction of vesicles to avoid massive computation; instead, $q$ is approximated as a linear interpolation between the values reported earlier for high LH2:RC and low LH2:RC chromatophore vesicles (*Şener et al., 2007, 2010, 2011*), namely between $q = 0.85$ and $q = 0.95$. For a vesicle containing $n_{\text{B}}$ cyt$bc_1$ dimers and $n_{\text{L}}$ LH1-RC dimers, the corresponding number of LH2 complexes $n_{\text{LH2}}(n_{\text{B}}, n_{\text{L}})$ is estimated by the excluded surface resulting from changes in $n_{\text{B}}$ and $n_{\text{L}}$ with respect to the reference vesicle in *Figure 1*. The corresponding quantum yield is estimated according to

$$q = 0.91 + 0.0152\,(s_0 - s) \tag{10}$$

chosen to reproduce the correct value of $q$ for the reference vesicle (*Figure 1*) as well as for the low LH2 limit (*Sener et al., 2011*); here $s = n_{LH2}/(2n_L)$ is the LH2:RC stoichiometry, which for the reference vesicle equals $s_0 = 2.625$; a lower limit of $q = 0.85$ is imposed to account for high LH2:RC vesicles (*Şener et al., 2007*), where the linear interpolation breaks down.

Validity of the generalized Förster formulation, thus outlined, has been demonstrated by excitation-transfer calculations employing the so-called hierarchy equation of motion formalism of stochastic quantum mechanics (*Ishizaki and Fleming, 2009b*; *Strümpfer and Schulten, 2012b*); the calculations show that photoexcitation of chromatophore BChls relaxes into a Boltzmann occupancy of exciton states within approximately 1 ps, i.e., faster than inter-complex transfer that takes 3–5 ps (*Hess et al., 1995*; *Strümpfer and Schulten, 2012b*). Accordingly, the assumption underlying generalized Förster theory, namely that transfer occurs from a thermally relaxed distribution of exciton states, holds in good approximation.

The final step in stage I of energy conversion is the formation of quinol from quinone at the RC. The respective formation rate can be expressed

$$k_{Q \to QH_2}(I) = \frac{1}{2} I q p_{RC}(I), \tag{11}$$

where the prefactor $\frac{1}{2}$ accounts for every quinol requiring two electron transfer events at the RC. Here $q$ is the quantum yield given by *Equation (8)* and $p_{RC}(I)$ is the probability for the RC to hold a quinone Q or a semiquinone QH, in either case the RC being ready to accept and convert an electronic excitation. The probability $p_{RC}(I)$ decreases with increasing $I$, since the quinone/quinol pool becomes quinol rich/quinone poor at increasing light intensities, due in part to coupling (*Figure 5*) to chromatophore redox factors, succinate dehydrogenase, NADPH dehydrogenase, and cytochrome $c$ oxidase (*Klamt et al., 2008*). As the quinone/quinol ratio decreases, it becomes less likely for RC to have a quinone/semiquinone available for electron transfer. The stated change in the quinone/quinol pool is crucial for energy conversion control of the chromatophore and comes about through the proton motive force, generated through light harvesting, inducing in the redox factors redox generation of products along with quinone/quinol conversion. The light-condition dependency of the quinone/quinol pool is described in the present model heuristically as explained below in *Equation (15–19)*.

Under the assumed steady state conditions, the rate $k_{Q \to QH_2}(I)$, i.e., the rate at which RCs release QH$_2$ as given by *Equation (11)*, is equal to the rate at which RCs bind fresh quinones. Accordingly holds

$$\frac{1}{2} I q p_{RC}(I) = \left[ n_{RC} \left( 1 - p_{RC}(I) \right) \right] / \tau_{RC}(I) \quad , \tag{12}$$

where $n_{RC} = 2 n_L$ is the number of RCs in the chromatophore (24 for the vesicle shown in *Figure 1*), $1 - p_{RC}(I)$ is the fraction of RCs ready to bind a fresh Q, $\tau_{RC}(I)$ is the mean time needed for a RC to become available for binding a new Q after it had just accepted a Q (*Remy and Gerwert, 2003*). Below, we refer to $\tau_{RC}(I)$ as the cycling time.

The probability $p_{RC}(I)$ is assumed, for convenience, to be uniform across all RCs rather than to vary between RCs due to inhomogeneities in the redox state of the quinone/quinol pool. This assumption is strictly valid only when the mixing time of quinols and quinones in the vesicle lipid phase is shorter than the time scales associated with the rates in *Equation (1)*. The spatial inhomogeneity of $p_{RC}(I)$ can be determined only through the simulation of the diffusive processes in the chromatophore, which is currently prohibitive.

It had been suggested in (*Geyer and Helms, 2006*) that the primary rate-limiting step in the chromatophore is quinol turnover at cyt$bc_1$ rather than cytochrome $c_2$ diffusion; in (*Geyer and Helms, 2006*) it had been estimated that each cytochrome $c_2$ is capable of approximately 80 electron transfers per second and that three cytochrome $c_2$'s per vesicle are sufficient to saturate the turnover capacity of an ATP synthase. A chromatophore vesicle is expected to feature 10–20 cytochrome $c_2$ molecules (*Geyer and Helms, 2006*; *Cartron et al., 2014*), safely exceeding the necessary number

needed for saturation. Therefore, cytochrome $c_2$ kinetics should not be rate limiting for energy conversion in the chromatophore.

Using **Equation (12)**, $p_{\mathrm{RC}}(I)$ can be expressed in terms of $\tau_{\mathrm{RC}}(I)$, namely,

$$p_{\mathrm{RC}}(I) = \left(1 + \frac{1}{2} I q \tau_{\mathrm{RC}}(I) \frac{1}{n_{\mathrm{RC}}}\right)^{-1}. \tag{13}$$

According to **Equations (11) and (13)**, $\tau_{\mathrm{RC}}(I)$ needs to be determined in order to compute the rate $k_{\mathrm{Q} \to \mathrm{QH}_2}(I)$ or, equivalently, $k_{\mathrm{ATP}}(I)$.

## Stage II: Diffusion of charge carriers and estimate of cycling time $\tau_{\mathrm{RC}}(I)$

The cycling time, $\tau_{\mathrm{RC}}(I)$, arising in **Equations (12,13)**, depends on light intensity. The cycling time is related to the quinone/quinol stoichiometry, i.e., the redox state of the quinone/quinol pool: the fewer quinones are present, the longer is the cycling time. The redox state is affected by not only RC and cyt$bc_1$ reactions, but also by transmembrane enzymes succinate dehydrogenase and NADH dehydrogenase (**Figure 5**). The low-light and high-light limits for the cycling time, $\tau_{\mathrm{RC}}$, employed below are based on experimental observation (**Woronowicz et al., 2011b**, **2011a**; **Crofts, 2004**) instead of direct computation; the reported values of $\tau_{\mathrm{RC}}$ implicitly combine the redox reactions of all enzymes interacting with the quinone/quinol pool, including NADH dehydrogenase.

In a stationary state, the rate $k_{\mathrm{ATP}}(I)$ of ATP synthesis is equal to the rate $k_{\mathrm{Q} \to \mathrm{QH}_2}(I)$ as stated in **Equation (1)**, which according to **Equations (11) and (13)** can be expressed through the cycling time, $\tau_{\mathrm{RC}}(I)$. The condition of equilibrium assumed here might not be valid for rapidly fluctuating light intensities, where spatial inhomogeneities of the vesicle and the quinone/quinol pool are expected to play a nontrivial role on the cycling time.

The low-light limit, $\tau_L$, of the cycling time, $\tau_{\mathrm{RC}}(I)$, is observed to range from 0.7 ms for the membrane of an LH2-minus mutant to about 3 ms for the LH2-rich chromatophores adapted to low-light growth conditions (**Woronowicz et al., 2011b**, **2011a**). In the following, we assume $\tau_L = 3\,\mathrm{ms}$ for the low-light growth vesicle shown in **Figure 1**.

At the high-light limit, the immediate vicinity of a RC contains mostly quinols and the replacement of the converted quinones at the RC becomes rate limited by the turnover at cyt$bc_1$ (**Woronowicz et al., 2011b**, **2011a**). The high-light limit, $\tau_H$, of the cycling time, $\tau_{RC}(I)$, can be estimated by considering the total turn-over rate at all RCs, namely $n_{\mathrm{RC}} \tau_H^{-1}$. In the stationary high $I$ regime, this rate must be equal to the quinol turnover rate at all cyt$bc_1$, namely $n_B \tau_B^{-1}$, i.e., it holds

$$n_B \tau_B^{-1} = n_{\mathrm{RC}} \tau_H^{-1}, \tag{14}$$

where $n_B$ is the number of cyt$bc_1$ dimers (4 for the vesicle shown in **Figure 1**) and $\tau_B = 25$ ms is the quinol turnover time at a cyt$bc_1$ (**Crofts, 2004**).

## Rate limitation of energy conversion by cyt$bc_1$

The estimate of the cycling time, $\tau_{RC}(I)$, given below is based on the observation that energy conversion in the chromatophore is rate limited by quinol turnover at cyt$bc_1$ (**Lavergne et al., 2009**; **Geyer et al., 2010**). This rate limitation follows directly from a comparison of turnover capacities, i.e., maximal turnover rates, at each key protein. The turnover rates of proteins are, in general, a function of the chromatophore conditions such as light intensity and redox states. The rate limiting components of the chromatophore can be identified by comparing the maximal values of the turnover rates, i.e., the turnover capacities, at each key protein, namely the quinol turnover capacity at cyt$bc_1$, the quinol generation capacity at RC and the proton utilization capacity at ATP synthase.

As an illustration of the cyt$bc_1$-limited kinetics, we first compare quinol turnover capacities at the cyt$bc_1$ and the RC. At a light intensity equivalent to 5% of full sunlight, i.e., 50 W/m$^2$, a chromatophore vesicle absorbs $I$=1860 photons/s (estimated from the functional absorption cross-section of a chromatophore given in [**Woronowicz et al., 2011a**]), corresponding to a quinol turnover capacity at the RCs of $\frac{1}{2} I q =846$ s$^{-1}$. In contrast, the quinol turnover capacity at all cyt$bc_1$s, given by **Equation (14)**, is equal to $n_B \tau_B^{-1} = 160\,\mathrm{s}^{-1}$. Hence, already at 5% of full sunlight the quinol production capacity at the RCs exceeds the total quinol turnover capacity at cyt$bc_1$ by more than five-fold. Consequently, under steady-state conditions the quinol production at the RCs at this illumination

becomes limited by quinol turnover at cyt$bc_1$. The onset of saturation of the energy conversion rate arising from rate limitation due to cyt$bc_1$ at low light intensities is evident in *Figure 2A*.

Next, we compare the proton turnover capacities at the cyt$bc_1$ and the ATP synthase. The maximal proton turnover capacity at cyt$bc_1$ for the vesicle shown in *Figure 1* is $4 \times n_B \tau_B^{-1} = 640$ s$^{-1}$. In comparison, total proton utilization capacity of ATP synthases is $4 \times 2 \times 270 = 2160$ s$^{-1}$, estimated based on the reported ATP synthase turnover capacity of 270 ATP molecules/s (*Etzold et al., 1997*) with four protons utilized per ATP and 2 ATP synthases present in the vesicle shown in *Figure 1*. Thus, the proton utilization capacity at ATP synthases exceeds the proton turnover capacity at cyt$bc_1$s by more than threefold. In summary, of the three potential kinetic bottlenecks in the chromatophore, cyt$bc_1$, RC, and ATP synthase, the lowest total turnover capacity is displayed by cyt$bc_1$.

## Light intensity dependence of the cycling time at RC

The $I$-dependence of the cycling time, $\tau_{RC}(I)$, needed to evaluate *Equation (13)*, is approximated in terms of the relative populations of a two state system, the two states corresponding to the low-light and high-light limits

$$\tau_{RC}(I) = c_L(I)\tau_L + c_H(I)\tau_H, \tag{15}$$

where $c_L(I)$ and $c_H(I)$ are the probabilities that quinol turnover follows the low-light (limited to RC vicinity) or high-light (cyt$bc_1$-limited) kinetics, respectively; it holds

$$c_L(I) + c_H(I) = 1. \tag{16}$$

The high-light limit of the cycling time, $\tau_H$, can be expressed using *Equation (14)*,

$$\tau_H = \frac{n_{RC}}{n_B}\tau_B. \tag{17}$$

Since low-light and high-light limits are actually the extremes of a gradual behavior, the assumption of a two-state system appears rather drastic. However, at low light levels $\tau_L$ is diffusion controlled and amounts to the first passage time of the quinone to the RC, while $\tau_H$ is determined by cyt$bc_1$ turnover, not diffusion. As a result one expects a distinct transition between $\tau_L$ and $\tau_H$ at some light intensity $I$ corresponding to the saturation of the rate-limiting process in the chromatophore. From *Equation (15,16)* follows

$$\tau_{RC}(I) = \tau_L + (\tau_H - \tau_L)(1 - c_L(I)). \tag{18}$$

The description of the cycling time according to *Equation (18)* is heuristic only. Future studies need to account for the time-dependent spatial inhomegenity of the quinone/quinol pool by explicitly modeling the diffusion processes and redox states in the chromatophore.

The population of photosynthetic states with respect to light intensity is typically governed by a Poisson distribution in terms of the utilization rate of excitations (*Mauzerall, 1986*; *Peterson et al., 1987*). In order to express the relative population of the low-light state, $c_L(I)$, we observe that charge separated states are created at the RCs with the rate $Iq$ and that the characteristic time for electron turnover at all cyt$bc_1$s is given by $1/(2 \times n_B \tau_B^{-1})$. The probability that no charge separation events occur during this time, i.e., the probability that the system remains in the low-light state, is given by the zero-event Poisson distribution, employed typically in describing the light intensity-dependence of photoproduct yield (*Mauzerall, 1986*; *Peterson et al., 1987*). According to this description holds, $c_L(I) = \exp(-\frac{1}{2}Iq/B)$, where $B = 2 \times n_B \tau_B^{-1}$ is the total turnover capacity of cyt$bc_1$s, which along with *Equation (18)* permits an estimate of the cycling time $\tau_{RC}(I)$, namely,

$$\tau_{RC}(I) = \tau_L + (\tau_H - \tau_L)\left(1 - e^{\frac{1}{2}IqB^{-1}}\right). \tag{19}$$

*Equation (19)*, when substituted into *Equations (11) and (13)*, permits an estimate of the quinol turnover rate $k_{Q \to QH_2}(I)$, employed below for the computation of the ATP synthesis rate.

## Stage III: ATP synthesis

As the last step of energy conversion, the proton gradient, generated at cyt$bc_1$ through quinol $\rightarrow$ quinone conversion, is utilized by ATP synthase for the production of ATP. The ATP turnover rate of the vesicle, $k_{\mathrm{ATP}}(I)$, under stationary conditions is equal to $k_{\mathrm{Q}\rightarrow\mathrm{QH}_2}(I)$ given by *Equation (11)*. This equality is based on the assumption that ATP synthase of *Rba. sphaeroides* has an $F_o$ of 12 c-subunits such that four $H^+$ conducted through the $F_o$-ring of ATP synthase lead to a 120° rotation of the stalk in the $F_1$ part and, thereby, to synthesis of one ATP. Currently, the structure of the ATP synthase of *Rba. sphaeroides* and the corresponding number of c-subunits is not known experimentally. If the $F_o$ oligomer were to feature, e.g., 11 or 10 subunits instead, this structural detail would proportionally affect the number of protons required for the rotation of the stalk and subsequent synthesis of each ATP and, therefore, directly influence the estimated energy conversion efficiency of the chromatophore, by 11% or 20%, respectively.

Combining *Equation (1),(11),(13)*, the ATP turnover rate can be expressed

$$k_{\mathrm{ATP}}(I) = \frac{1}{2}Iq\left(1 + \frac{1}{2}Iq\,\tau_{\mathrm{RC}}(I)\frac{1}{n_{\mathrm{RC}}}\right)^{-1},\tag{20}$$

where the cycling time at the RC, $\tau_{\mathrm{RC}}(I)$, is given by *Equation (19)*.

The overall energy conversion efficiency of the chromatophore, $\eta_{\mathrm{ATP}}(I)$, can be defined as the ratio of formation rate of energy in the form of ADP→ATP synthesis to the total absorption rate of photon energy

$$\eta_{\mathrm{ATP}}(I) = \frac{E_{\mathrm{ATP}}\,k_{\mathrm{ATP}}(I)}{E_\gamma I},\tag{21}$$

where $E_{\mathrm{ATP}} = 4197\sim\mathrm{cm}^{-1}$ is the ATP hydrolysis energy in the cell (*Berg et al., 2011*) and $E_\gamma$ is the average energy of an absorbed photon, taken to be the photon energy at 850 nm (11765 $\mathrm{cm}^{-1}$). Not all the energy of an absorbed photon, $E_\gamma$, is available for energy harvesting. The fraction of $E_\gamma$ available for conversion into chemical energy is the so-called Carnot yield (*Lavergne, 2009*) described by comparing photochemical energy conversion to the function of a heat engine. This limitation in photochemical energy conversion establishes a theoretical upper limit for photosynthetic energy conversion at broad daylight of approximately 0.7 (*Lavergne, 2009*).

The determination of the energy conversion efficiency, $\eta_{\mathrm{ATP}}(I)$, computed through *Equation (21)*, has the shortcoming that the ATP hydrolysis energy, $E_{\mathrm{ATP}}$, depends, in principle, on the ADP, ATP, and $H^+$ concentrations in the cytoplasm, which are not modeled explicitly. Nevertheless, *Equation (21)* permits a comparison with similar measures of efficiency reported for other photosynthetic or photovoltaic systems (*Blankenship et al., 2011*).

## Supplementary material
### Computation of quantum yield and table of BChl properties

In the following, we describe the computation of the quantum yield, $q$, given by *Equation (8)*, in section 2.2 of the main text. The quantum yield is central to the overall efficiency of the chromatophore as it accounts for the efficiency of the primary subsystem, the light harvesting apparatus. The constants employed in the computation of the quantum yield are listed below in *Table 1*. We adopt the same values as in similar computations reported by us in (*Sener et al., 2010, 2007*). For the sake of clarity, this section repeats some information in the main text, as indicated.

In order to describe the electronic excitation transfer in the chromatophore and determine the quantum yield, $q$, the matrix of transfer rates, $k_{IJ}$, between BChl clusters, $I,J$, needs to be constructed, which in turn depends on the effective Hamiltonian, $H_I$, for each BChl cluster, $I$. In the following, the indices, $I,J$, label the BChl clusters listed in the previous section: for the vesicle shown in *Figure 1*, based on (*Cartron et al., 2014*), there are 63 LH2 B850 BChl clusters, 24 LH1 B875 BChl clusters, and 24 RC BChl clusters. As mentioned in the previous section, LH2 B800 BChls do not form excitonically coupled states, transferring excitation energy, after light absorption, immediately to LH2 B850 BChls.

The effective Hamiltonian $H^I$ of each BChl cluster $I$ is given, according to (*Strümpfer et al., 2012*), by *Equation (2)* in the main text, namely

**Table 1.** Constants employed in the computation of the quantum yield.

| Symbol | Value* | Description |
|---|---|---|
| $\epsilon_1^{LH2}\epsilon_2^{LH2}$ | 12,459 cm$^{-1}$ 12,625 cm$^{-1}$ | BChl site energies for alternating LH2 B850 BChls, used in **Equation (S2)** |
| $\epsilon^{LH1}$ | 12,344 cm$^{-1}$ | BChl site energies for LH1 B875 BChls, used in **Equation (S2)** |
| $\epsilon_1^{RC}\epsilon_2^{RC}$ | 12,092 cm$^{-1}$ 12,581 cm$^{-1}$ | BChl site energies for RC special pair and accessory BChls, used in **Equation (S2)** |
| $V_1^{LH2}$ | 363 cm$^{-1}$ | nearest neighbor BChl-BChl coupling for LH2 B850 BChls within the same $\alpha\beta$ dimer, used for $V_{ij}^I$ values in **Equation (S3)** |
| $V_2^{LH2}$ | 320 cm$^{-1}$ | nearest neighbor BChl-BChl coupling for LH2 B850 BChls across neighboring $\alpha\beta$ dimers, used for $V_{ij}^I$ values in **Equation (S3)** |
| $V_1^{LH1}$ | 806 cm$^{-1}$ | nearest neighbor BChl-BChl coupling for LH1 B875 BChls within the same $\alpha\beta$ dimer, used for $V_{ij}^I$ values in **Equation (S3)** |
| $V_2^{LH1}$ | 377 cm$^{-1}$ | nearest neighbor BChl-BChl coupling for LH1 B875 BChls across neighboring $\alpha\beta$ dimers, used for $V_{ij}^I$ values in **Equation (S3)** |
| $V^{RC}$ | 500 cm$^{-1}$ | Coupling between RC special pair BChls, used in **Equation (S3)** |
| $C$ | 348,000 Å$^3$ cm$^{-1}$ | coupling constant for transition dipole interactions between non-nearest neighbor LH2 BChls, used in **Equation (S4)** |
| $\sigma^{LH1}$ | 235 cm$^{-1}$ | linewidth of LH1 exciton states, assumed uniform, used for $\sigma_A$ in **Equation (S8)** |
| $\sigma^{LH2}$ | 188 cm$^{-1}$ | linewidth of LH2 exciton states, assumed uniform, used for $\sigma_A$ in **Equation (S8)** |
| $k_{LH1,RC}$ | (35 ps)$^{-1}$ | excitation transfer rate from LH1 B875 BChls to RC, used for corresponding $k_{IJ}$ values in in **Equation (S10)** |
| $k_{RC,LH1}$ | (8 ps)$^{-1}$ | excitation transfer rate from RC to LH1 B875 BChls, used for corresponding $k_{IJ}$ values in in **Equation (S10)** |
| $k_{diss}$ | (1 ns)$^{-1}$ | excitation decay rate due to internal conversion, used in **Equation (S10)** |
| $k_{CS}$ | (3 ps)$^{-1}$ | charge separation rate at reaction center, used in **Equation (S10)** |

*: energy units given in wavenumbers (1 eV = 8066 cm$^{-1}$).

$$H^I = \sum_{i=1}^{N_I} E_i^I |i\rangle\langle i| + \sum_{i>j>0}^{N_I} V_{ij}^I (|i\rangle\langle j| + |j\rangle\langle i|), \tag{S1}$$

where $N_I$ is the number of BChls in cluster $I$. The site energies $E_i^I$ in **Equation (S1)** are

$$E_i^I = \begin{cases} \epsilon_{1,2}^{LH2}, & I : \text{LH2 B850,} \\ \epsilon^{LH1}, & I : \text{LH1 B875,} \\ \epsilon_{1,2}^{RC}, & I : \text{RC,} \end{cases} \tag{S2}$$

with energy constants as listed in **Table 1** for corresponding BChl groups, chosen to reproduce corresponding absorption peaks. The couplings $V_{ij}^I$ in **Equation (S1)** are determined through the point-dipole approximation as described below with the exception of nearest neighbor couplings of the LH2 B850 and LH1 B875 BChl clusters and the RC special pair coupling,

$$V_i^I = \begin{cases} V_{1,2}^{LH2}, & I : \text{LH2 B850, nearest neighbor,} \\ V_{1,2}^{LH1}, & I : \text{LH1 B875, nearest neighbor,} \\ V^{RC}, & I : \text{RC, special pair,} \end{cases} \tag{S3}$$

which are instead taken from (**Damjanović et al., 2000**; **Şener and Schulten, 2002**) following quantum chemistry computations reported in (**Tretiak et al., 2000**) and listed in **Table 1**.

For non-nearest neighbor BChls, the coupling $V_{ij}^I$ in **Equation (S1)** is computed according to the point-dipole approximation (**Ritz et al., 2001**; **Sener et al., 2011**) (**Equation (3)** in the main text)

$$V_{ij} = C\left( \frac{\hat{\mathbf{d}}_i \cdot \hat{\mathbf{d}}_j}{r_{ij}^3} - 3\frac{\left(\hat{\mathbf{d}}_i \cdot \mathbf{r}_{ij}\right)\left(\hat{\mathbf{d}}_j \cdot \mathbf{r}_{ij}\right)}{r_{ij}^5} \right), \tag{S4}$$

where $\hat{\mathbf{d}}_i$ is the transition dipole moment unit vector of pigment $i$, $\mathbf{r}_{ij}$ is the vector joining pigments $i$ and $j$, in BChl cluster $I$; the coupling constant $C$ (**Şener and Schulten, 2002**) is listed in **Table 1**. The

transition dipole moment unit vector $\hat{\mathbf{d}}_k$ for BChl $k$ is determined from the coordinates listed in the previous section according to

$$\hat{\mathbf{d}}_k = \frac{(\mathbf{r}_k^D - \mathbf{r}_k^B)}{|\mathbf{r}_k^D - \mathbf{r}_k^B|}, \tag{S5}$$

where $\mathbf{r}_k^B$ and $\mathbf{r}_k^D$ are the positions of atoms NB and ND of BChl $k$; the transition dipole moment is centered at the position, $\mathbf{r}_k^M$, of the MG atom of BChl $k$, as labeled in the PDB files.

Based on the effective Hamiltonians, *Equation (S1)*, thus constructed, the rate of excitation transfer between a donor complex $I$ and an acceptor complex $J$, $k_{IJ}$, can be calculated according to the so-called modified Förster theory (*Ritz et al., 2001*; *Şener et al., 2007, 2011*) given by *Equation (4)* in the main text, namely through

$$k_{IJ} = \frac{2\pi}{\hbar} \sum_{\mu \in I} \sum_{\nu \in J} p_\mu^I |(\mu|H_{IJ}|\nu)|^2 J_{\mu\nu}, \tag{S6}$$

where $(H_{IJ})_{ij}$ is the matrix of interactions between the excited states of pigments $i$ and $j$ in complexes $I$ and $J$, computed according to *Equation (S4)* and $(\mu|H_{IJ}|\nu)$ are the couplings $(H_{IJ})_{ij}$ in the basis of the eigenstates $|\mu\rangle$ and $|\nu\rangle$ of Hamiltonians $H_I$ and $H_J$, respectively; $p_\mu^I = e^{-\beta\epsilon_\mu}/\sum_{\gamma \in I} e^{-\beta\epsilon_\gamma}$ are Boltzman weights for the eigenstates $H_I|\mu\rangle = \epsilon_\mu|\mu\rangle$, where $\epsilon_\mu$ are the exciton energies defined as the eigenvalues of the Hamiltonian, $H_I$, given in *Equation (S1)*. The overlap integrals $J_{\mu\nu}$ are described according to *Equation (5)* in the main text)

$$J_{\mu\nu} = \int dE\, S_\mu^D(E) S_\nu^A(E). \tag{S7}$$

$J_{\mu\nu}$ is the spectral overlap between donor exciton state $|\mu\rangle$ and acceptor exciton state $|\nu\rangle$. The donor and acceptor lineshapes, $S_\mu^D(E)$ and $S_\nu^A(E)$, used in the calculation of $J_{\mu\nu}$, are approximated by normalized Gaussians

$$S_\mu^A(E) = \frac{1}{\sqrt{2\pi}\sigma_A} \exp\left[-\left(\frac{E-\epsilon_\mu}{\sigma_A}\right)^2\right], \tag{S8}$$

$$S_\mu^D(E) = S_\mu^A(E-S), \tag{S9}$$

$\sigma_A$ is the linewidth of excitons assumed to be uniform across all states (*Jimenez et al., 1997*) and is given in *Table 1* as $\sigma^{LH1}$ and $\sigma^{LH2}$; $S$ denotes the spectral shift between donor and acceptor spectra (*Small, 1995*; *Damjanović et al., 2000*).

The transfer rates, $k_{IJ}$, in *Equation (S6)* are negligible for any non-neighboring BChl clusters, $I, J$, and are taken to be zero in those cases. The transfer rates between the B875 and the RC BChls of a LH1-RC complex needs to be determined only once, since the relative pigment geometry is identical within each LH1-RC complex. Accordingly, for the LH1→RC and RC→LH1 transfer rates, the values $(35\ \text{ps})^{-1}$ and $(8\ \text{ps})^{-1}$ are assumed, respectively (*Sener et al., 2010, 2009*), in *Table 1*; the LH1→RC value was chosen in (*Sener et al., 2010*) to match the observed excitation lifetime in the RC-LH1 complex of 50 ps; the value of the LH1→RC transition rate, computed according to *Equation (S6)*, was reported to result in an overestimate of the excitation lifetime (*Sener et al., 2009*). The remaining transfer rates, $k_{IJ}$, namely between neighboring LH2 B850 and LH1 B875 BChl clusters are determined according to *Equation (S6)*.

Exciton migration in the chromatophore is governed by the rate matrix for inter-complex exciton transfers, $\mathcal{K}_{IJ}$ (*Sener et al., 2010, 2007*) *Equation (7)* in the main text)

$$(\mathcal{K})_{IJ} = k_{JI} - \delta_{IJ}\left(\sum_M k_{IM} + k_{\text{diss}} + k_{\text{CS}}\delta_{I,\text{RC}}\right), \tag{S10}$$

where the dissipation rate $k_{\text{diss}}$ and the charge separation rate $k_{\text{CS}}$ are listed in *Table 1*. The quantum yield, $q$, is finally expressed in terms of $\mathcal{K}_{IJ}$ (*Şener et al., 2007, 2011*) according to *Equation (8)* in the main text, namely through

$$q = -k_{CS} \left( \mathbf{1}_{RC} \right)^T \cdot \mathcal{K}^{-1} \cdot \mathbf{P}(0), \tag{S11}$$

where $(\mathbf{P}(0))_I = N_I / (\sum_J N_J)$, corresponds to a uniform probablity for initial excitation. The quantum yield is not strongly dependent on the choice of the initial state $(\mathbf{P}(0))$ (*Şener et al., 2007*).

The quantum yield, $q$, is computed by substituting the BChl atom coordinates, as listed in the caption of *Supplementary file 1*, into *Equations (S4 and S5)* to determine the couplings, $V_{ij}$, and subsequently the matrices $k_{IJ}$ and $\mathcal{K}_{IJ}$ according to *Equations (S6 and S10)*, substituted finally into *Equation (S11)*. The quantum yield of the chromophore shown in *Figure 1*, thus computed, is 0.91.

## Acknowledgements

The reviewers are thanked for extensive suggestions to improve the manuscript, particularly regarding comments on processes in the chromatophore beyond the light harvesting-cyt$bc_1$-ATP synthase components. The authors would like to also thank Antony Crofts, Robert Niederman, and Donald Bryant for insightful discussions on chromatophore function. This work was supported by the National Science Foundation (MCB1157615 and PHY0822613) (to KS), the National Institutes of Health (NIH 9P41GM104601) (to KS). CNH acknowledges research grant BB/M000265/1 from the Biotechnology and Biological Sciences Research Council (UK). CNH was also supported by an Advanced Award 338895 from the European Research Council. This research used resources of the Oak Ridge Leadership Computing Facility at the Oak Ridge National Laboratory, which is supported by the Office of Science of the U.S. Department of Energy under Contract No. DE-AC05-00OR22725. The study reported was funded also by the Photosynthetic Antenna Research Center (PARC), an Energy Frontier Research Center supported by the US Department of Energy, Office of Science, and Office of Basic Energy Sciences under Award Number DE-SC0001035 (to CNH and KS). The molecular image in *Figure 1* was generated with VMD (*Humphrey et al., 1996*).

## Additional information

### Funding

| Funder | Grant reference number | Author |
|---|---|---|
| Biotechnology and Biological Sciences Research Council | BB/M000265/1 | C Neil Hunter |
| U.S. Department of Energy | DE-SC0001035 | C Neil Hunter<br>Klaus Schulten |
| European Research Council | 338895 | C Neil Hunter |
| National Science Foundation | PHY0822613 | Klaus Schulten |
| National Institutes of Health | NIH 9P41GM104601 | Klaus Schulten |
| National Science Foundation | MCB1157615 | Klaus Schulten |

The funders had no role in study design, data collection and interpretation, or the decision to submit the work for publication.

### Author contributions

MS, JS, AS, CNH, KS, Conception and design, Acquisition of data, Analysis and interpretation of data, Drafting or revising the article

### Author ORCIDs

Klaus Schulten, http://orcid.org/0000-0001-7192-9632

# Additional files

## Supplementary files

• Supplementary file 1. BChl groups of the chromatophore and the corresponding transition dipole moments needed to determine the effective Hamiltonian: LH2 complexes. (file: Vesicle30BCLdipoleLH2.pdb) Here we provide the transition dipole moments of all 2577 chromatophore BChls defining the effective Hamiltonian in *Equations (2,3)* of the main text, needed for evaluation of the excitation transfer rates through *Equation (4)*. The reader is referred to (*Cartron et al., 2014*; *Şener et al., 2007*, *2010*) for the construction of the underlying chromatophore structural model. The three PDB files listed here contain the coordinates for the MG, NB, ND atoms of the BChls in the chromatophore model (Cartron et al., 2014) that define the transition dipole moment unit vectors as discussed in the text and given according to *Equation (S1)* below. These coordinate files correspond to the following protein complexes: *LH2* : [Vesicle30BCLdipoleLH2.pdb], *dimeric RC-LH1* : [Vesicle30BCLdipoleLH1RCdimer.pdb], *monomeric RC-LH1* : [Vesicle30BCLdipoleLH1RCdimer.pdb], where coordinates of multiple complexes of the same type are concatenated into one file for each complex type.The BChls of the chromatophore can be divided into the following groups: (i) B800 BChls of LH2 : labeled as [resid=307] in the file [Vesicle30BCLdipoleLH2.pdb] (resid is used here as an abbreviation for residue sequence number in the PDB file format); named after the absorption peak of 800 nm; (ii) B850 BChls of LH2 : labeled as [resid=301, 302] in the file [Vesicle30BCLdipoleLH2.pdb]; named after the absorption peak of 850 nm; (iii) B875 BChls of LH1-RC (both monomer and dimer) : labeled as [resid=100-155] in the files [Vesicle30BCLdipoleLH1RCdimer.pdb] and [Vesicle30BCLdipoleLH1RCmonomer.pdb]; named after the absorption peak of 875 nm; (iv) reaction center (RC) BChls of LH1-RC (both monomer and dimer) : labeled as [resid=301, 302, 303, 304] in the files [Vesicle30BCLdipoleLH1RCdimer.pdb] and [Vesicle30BCLdipoleLH1RCmonomer.pdb]; the so-called special pair BChls where electron transport is initiated are labeled by [resid = 302, 303]. Of the aforementioned BChl clusters, the three, namely, LH2 B850, LH1 B875, and RC BChls, form strongly coupled excitonic states; the LH2 B800 BChls do not share excitation energy between themselves, transfering it rapidly (within 0.5 ps) (*Şener et al., 2007*; *Ritz et al., 2001*) to the B850 ring of the same LH2. The theory of excitation transfer between the BChl clusters listed above is described in Supplementary materials.

• Supplementary file 2. BChl groups of the chromatophore and the corresponding transition dipole moments needed to determine the effective Hamiltonian: RC-LH1 dimer complexes. (file: Vesicle30BCLdipoleLH1RCdimer.pdb) As explained for *Supplementary file 1*, but for BChls belonging to RC-LH1 dimers.

• Supplementary file 3. BChl groups of the chromatophore and the corresponding transition dipole moments needed to determine the effective Hamiltonian: RC-LH1 monomer complexes. (file: Vesicle30BCLdipoleLH1RCmonomer.pdb) As explained for *Supplementary file 1*, but for BChls belonging to RC-LH1 monomers.

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
