## [Decision Letter]

Thank you for submitting your work entitled "Overall energy conversion efficiency of a photosynthetic vesicle" for peer review at *eLife*. Your submission has been favorably evaluated by Michael Marletta (Senior editor), a Reviewing editor, and three reviewers. We apologize for the delay in rendering a decision, but it was necessary to find multiple independent reviews on this manuscript.

The reviewers have discussed the reviews with one another and the Reviewing editor has drafted this decision to help you prepare a revised submission.

Summary:

The authors present a model of the *Rhodobacter sphaeroides* chromatophore. The model employs a detailed quantum mechanical description for the light harvesting and charge separation processes, and a kinetic description for the subsequent reactions. With this model, the solar energy capture efficiency is determined, as measured in terms of ATP production. Efficiencies are estimated to be on the order of 10%. The authors conclude that the vesicle stoichiometry is sub-optimal for steady-state ATP production and instead appears to be adapted to "robustness of function".

Essential revisions:

All three reviewers are overall positive, but raise a number of issues concerning the model, its limitations, and its interpretation. I find the issues raised by the referees to be relevant. Addressing them in a revision should result in a stronger paper. Following is a brief summary of their reports that emphasizes the main points as seen from my perspective.

Reviewer #1 challenges the conclusion that suboptimality of ATP production is associated with robustness of function. First, the reviewer points out that the build-up of a QH2 pool would only work if an equal amount of cytochrome *c_2_* were present (in essence because the electron cycle with the chromatophore is closed). Since the number of cyt c is much smaller, something seems to be amiss, either in the model or its interpretation. Second, the reviewer argues that evolution acts on the survival of the organism, not at the level of its chromatophores.

Reviewer #1 is also concerned about the absence of leak reactions in the model, in particular for electrons between quinol and the Rieske protein, and for protons through the chromatophore membrane.

Reviewer #2 is concerned that the model lacks physiological relevance, despite overall being positive and recognizing the value of a reduced model. In particular, the Reviewer notes that chromatophore operation is connected to NADH production. If this link were included, the reviewer argues that the buildup of the ubiquinol pool would not occur. At the least, the authors should discuss the more complex situation of chromatophore function in phototropic bacteria.

Concerning the model it is assumed that the F_o_ part of the ATP synthase has a c-ring with 12 subunits, but to the best of knowledge, this is only a guess. If so, it should be stated as such.

Additional major points are the precise definitions of efficiency metrics (Reviewer #1) and a discussion as to what the atomistic description of the chromatophore really adds. Moreover, all reviewers point to the need for a more thorough discussion of the relevant literature.

After the reviews were returned, the editors discussed your work in relation to the broader audience of *eLife*. We urge you to include language in the Discussion section that expands the scope of your presentation to include a description of how your finding may relate to other systems.

In light of the reviewers comments, changes will be necessary before the paper can be considered for publication. Following are their full reports. These should be consulted and addressed point by point.

Reviewer #1:

In this work the authors calculate the solar energy capture efficiency of a bacterial photosynthetic vesicle based on an explicit model for the structure of the photosynthetic apparatus. While a detailed quantum mechanical model is used for the light harvesting and charge separation process, for all the other subsequent reactions kinetic models are utilized. At solar intensities of 1-5% of full sunlight the energy conversion efficiency is calculated to range from 0.12 to 0.04. The vesicle stoichiometry appears to be sub-optimal for steady-state ATP production and appears to be adapted to "robustness of function".

In general this is a well-written paper, however there are some serious problems in the interpretation that the authors must account for.

1) The authors suggest that the suboptimality of the ATP-production of the *Rba. Sphaeroides* vesicle might play a role in the robustness of its function. According to their interpretation reduction of the Q-pool might play a role in storing the redox energy accumulated during short periods of exposure to high light. This would only work if there is an equal amount of cytochrome c present. However, there is one to two orders of magnitude more quinone than cytochrome c so where do the electrons come from that reduce the Q-pool?

2) Evolution did not act at the level of individual chromatophores but on the survival fitness of the intact organism.

3) The authors use efficiency values without specifying exactly what they mean. I presume that they have had a certain numerical value for "delta-Gp" in mind (i.e. the standard free energy of hydrolysis of ATP), and then calculate efficiency as the ratio of the caloric value of the photons over the free energy of the corresponding ATP hydrolysis. This should however be made explicit. Furthermore, they now and then shift back and forth from oxygenic to anoxygenic photosynthesis (e.g. Discussion, first paragraph), which is not proper because in oxygenic photosynthesis the majority of the energy conservation goes into redox energy (i.e. NADPH). Estimates that anoxygenic photosynthesis has a free-energy efficiency in the order of 10% have already been available for a long time (Hellingwerf, K. J., Crielaard, W., & Westerhoff, H. V. (1993). Comparison of retinal-based and chlorophyll-based photosynthesis: A biothermokinetic description of photochemical reaction centers. Modern trends in biothermokinetics (pp. 45-52) Springer).

4) Beyond the LH/RC itself, nowhere in the modelling has any leak-reaction taken into account. Two examples that would have been very relevant to take into account are: (a) the short-cut in electron transfer in the cytochrome *bc*_1_ complex that may occur if the second electron of quinol is also directly transferred to the Rieske protein and (b) passive proton leakage through the chromatophore membrane (which may be very much dependent of the protein composition of the vesicles, due to curvature effects). Related to this is the fact that the authors seem to suggest that proton transport merely leads to a pH change (see discussion on acidification), whereas presumably the energetic coupling between the proton pumping system and the ATP synthase is through a transmembrane electrical potential gradient.

5) What precisely does the atomistic description of the light harvesting and trapping process contribute to the calculation of the efficiency? Could the energy transfer and trapping process not be captured by a single effective rate constant that is a function of the fraction of open and closed traps and the ratio of LH2 to RC-LH1. See many earlier papers on the subject: Vredenberg and Duysens, Duysens CIBA foundation, Den Hollander et al., Trissl et al., Joliot cs etc.

Reviewer #2:

The manuscript by Sener and co-workers presents a model of a *Rhodobacter sphaeroides* chromatophore. The model describes how the illumination of a chromatophore suspension results in ATP production. The ultimate goal of the modeling is to understand how the ATP yield is coupled with the intensity of light. The main virtue of the model is that it couples the quantum physics of light absorption/procession with the chemistry of ATP synthesis. The model gives reasonable values of ATP yield in response to low light, which is physiological for these bacteria. Development of such a global model, which covers events over a time span of 14 orders of magnitude, required certain simplifications, which is understandable. Unfortunately, from the opinion of this reviewer, the simplifications preclude from making physiologically relevant conclusions from the modeling, at least, at the given stage.

Still the model itself is a great achievement and the manuscript deserves publication albeit after certain amendments.

1) Since the physiological relevance of the results obtained is questionable, the main result of this study seems to be the model itself. Therefore, the description of the model should be moved from the Materials and methods into the Results. Otherwise the manuscript is incomprehensive, taking into account that the format of *eLife* implies Materials and methods after Results. The Materials and methods section should then contain only technical/purely methodical points; supplementary materials, at least partly, could be also moved into Materials and methods.

2) The current simplified model lacks two important modules, which, hopefully, would be added in the future. These are the protonmotive force and reversible NADH dehydrogenase. The authors admit the absence of these modules, but did not consider the consequences of their absence in full extent. This should be done in the Discussion.

3) In the absence of these two modules, the authors conclude that "energy conversion in the chromatophore appears to be rate-limited primarily by cyt*bc_1_*" and that "cyt*bc_1_*-limited kinetics prevents an overproduction of protons at sustained high-light conditions,…thereby protecting the chromatophore against overacidification of its interior and assuring vesicle integrity at high-light conditions".

In real phototropic bacteria, the situation is more complex. Obviously, the function of a photosynthetic apparatus, such as bacterial chromatophore or a chloroplast thylakoid, is to produce ATP and NAD(P)H. In thylakoids, NAD(P)H molecules are produced via direct reduction of NAD(P)+ by the Photosystem I. In *Rb. sphaerides*, the redox potential of the RC is too high for NAD(P)H generation. Therefore, NAD(P)H is produced via reversing the NAD(P)H dehydrogenase. The reversion requires (1) high protonmotive force and (2) reduced ubiquinol pool. Therefore, the overreduction of the ubiquinol pool in *Rb. sphaeroides* is a precondition for the NAD(P)H production; it should increase the overall efficiency of the system. Furthermore, the overacidification of the interior is prevented not by the overreduction of the ubiquinone pool, but by the high sensitivity of the cyt*bc_1_*to the membrane potential. High voltage, needed to drive the NAD(P)+ reduction (see above), slows the turnover of the cyt*bc_1_*dramatically, essentially blocking its turnover at voltages above 150 mV. Hence at high, physiological values of protonmotive force the turnover of the cyt*bc_1_*is essentially limited by the proton discharge, specifically via the ATP synthase and the reverse NAD(P)H dehydrogenase. The operation of the cyt*bc_1_*close to equilibrium additionally increases the efficiency of energy conversion.

The reviewer admits that the data on synthesis of NAD(P)H in *Rb. sphaeroides* are too scarce to be used for verification of the model. The authors, accordingly, have tested their model by using more abundant data that were obtained with chromatophore suspensions in the absence of NAD(P)+, in experiments where the function of the RC, cyt*bc_1_*and/or ATP synthase were studied. And still, the discussion of physiological consequences from the given model should be truncated dramatically as misleading. The physiology could be discussed in further publications, after the membrane potential and NAD(P)H synthesis would be explicitly included in the model.

Instead, the Discussion section should include comparison of the given model with the earlier models of chromatophore systems (e.g. from the group of Helms) and the approaches used.

Reviewer #3:

The authors present an important theoretical investigation of the performance and limitations of the photosynthetic chromatophore complex of purple bacteria. This light-harvesting system is of great interest as a model system that contains numerous characteristic properties inherent to all natural light-harvesting systems. This work is a continuation of a long line of first-rate papers published by the authors on this topic, is extremely well written, and merits publication in its current form.

There appears to be a minor inconsistency in the units of intensity employed. In the second paragraph of section 4.2 the authors define light intensity, I, in units of [photons absorbed per second], whereas light intensity is often defined in units of [photons absorbed per meter squared per second]. As defined, this intensity is in fact a power, which is also called a 'radiant flux' in radiometry. This in contrast with the more conventional definition of intensity, in the caption of Figure 2, where units of [Watts per meter squared] are used. Clarification of these different definitions would be useful.

In the third paragraph of section 4.2, the authors switch to defining 'I' as 'illumination,' which may further confuse the reader. Ideally 'radiant flux' or 'absorbed light power' or similar terminology would be employed throughout, or alternatively 'illumination' would be defined.

The explanation provided in the fourth and fifth paragraphs of section 2.2 for the suboptimal (nB, nRC) values in low-light adapted vesicles is appealing, but it does not appear to be supported by calculations. Despite the fact that the model only applies in steady-state, is it possible to perform a simple simulation to support this interpretation, even qualitatively?

The choice of initial condition assumed in [Disp-formula equ9] may possibly be over-simplifying because each complex (LH2, LH1 and RC) does not necessarily have the same absorption spectrum, since at a given wavelength, the probability of absorption of a photon is not the same for each complex. In this scenario, the initial condition should then be wavelength dependent. While the authors state that the quantum yield is not strongly dependent on the choice of P(0), a specific example to demonstrate this would be helpful.

[Editors' note: further revisions were requested prior to acceptance, as described below.]

Thank you for resubmitting your work entitled "Overall energy conversion efficiency of a photosynthetic vesicle" for further consideration at *eLife*. Your revised article has been favorably evaluated by Michael Marletta (Senior editor), a Reviewing editor, and two reviewers.

On the basis of the referee reports on the revised manuscript, only relatively minor additional changes are required. Reviewer #2 is concerned that some issues have not been fully addressed in the revision. In the following I list the main points.

1) The reviewer maintains that "the model does not consider the production of NADH at all." Therefore, "no statements on the energy conversion efficiency could be done with the current model." The reviewer is concerned that the issue of NADH production and its impact on the model and its assessment are not discussed adequately, and advises "to read the primary literature on the mechanism of NADH production and provide references to this literature (currently they refer only to the paper of Klamt et al. where bacterial photosynthesis was modeled by using several dozen parameters that were more or less freely variable). The primary literature describes how the NADH-dehydrogenase makes NADH by driving the reverse (uphill) electron transfer from the ubiquinone pool to NAD+ with the help of membrane potential. This mechanism differs from the NAD(P)H production in green plants where NAD(P)+ is directly reduced by photoexcitated electrons in a reaction that is independent of membrane potential. The dependence of the light-driven NADH production on membrane potential, specific for *Rhodobacter* and other purple phototrophic bacteria, should have shaped the properties of their energy-converting machinery."

Recommended action: Whereas I do not see the lack of NADH production as a fundamental flaw in the model (after all, all models are incomplete), this point should be made clearer to the reader. Conclusions drawn from the model should be qualified accordingly. In particular, in the discussion of the energy conversion efficiency, relevant limitations of the model should be stated.

2) The reviewer also asks to point out clearly that "the experimental data that were used for testing the model were not obtained with chromatophores in vivo," but for "an artificial system – a suspension of chromatophores in a pH-buffer, in the presence of electron donors and redox mediators, but in the absence of physiological compounds such as NAD+/NADH (a Dutton/Crofts system)."

Recommended action: Point out the distinction between the in-vitro experiments and the more complex in-vivo situation.

3) The reviewer also argues that the effect of a transmembrane proton gradient, which is not included explicitly in the model, should be discussed. The reviewer emphasizes that the issue is not "proton migration from the *bc*_1_ complex to ATP synthase" but protonmotive force.

Recommended action: Discuss the issue of protonmotive force. Examine which conclusions would likely be impacted by the explicit treatment of PMF in the model, and qualify them accordingly. If none of the conclusions would be affected, state that the model is robust with respect to the neglect of explicit modeling of PMF.

4) The reviewer suggests adding a description of the model to the Results section.

---

## [Author Response]

*Reviewer #1:*

*In this work the authors calculate the solar energy capture efficiency of a bacterial photosynthetic vesicle based on an explicit model for the structure of the photosynthetic apparatus. While a detailed quantum mechanical model is used for the light harvesting and charge separation process, for all the other subsequent reactions kinetic models are utilized. At solar intensities of 1-5% of full sunlight the energy conversion efficiency is calculated to range from 0.12 to 0.04. The vesicle stoichiometry appears to be sub-optimal for steady-state ATP production and appears to be adapted to "robustness of function".*

*In general this is a well-written paper, however there are some serious problems in the interpretation that the authors must account for.*

*1) The authors suggest that the suboptimality of the ATP-production of the Rba. Sphaeroides vesicle might play a role in the robustness of its function. According to their interpretation reduction of the Q-pool might play a role in storing the redox energy accumulated during short periods of exposure to high light. This would only work if there is an equal amount of cytochrome c present. However, there is one to two orders of magnitude more quinone than cytochrome c so where do the electrons come from that reduce the Q-pool?*

The energy buffering capacity of the Q-pool, arising between light absorption and ATP synthesis, is limited, as the reviewer states, by the number of electrons available in the photosynthetic apparatus that can shift the quinone-quinol balance. However, the quinone-quinol balance can be shifted also through redox reactions coupling the chromatophore to the redox pools in the cell's cytoplasm by means of membrane-bound enzymes listed below and in the text. The current structural model (as presented in Figure 1) does not include such enzymes and, therefore, cannot account for sufficient buffering capacity of the Q-pool. As the storage of energy in case of light fluctuations is not an essential outcome of our study, we follow the reviewer's suggestion and removed from the manuscript the interpretation regarding the buffering effect (last sentence of the Abstract; Introduction; Results; Discussion; Materials and methods.

As the reviewer states, the redox buffering capacity of the Q-pool is dependent on the electrons available in the chromatophore for quinone-to-quinol reduction. When considering the chromatophore components shown in Figure 1, only a limited number of electrons are maximally available: 24 from RC, 20 from cyt. *c_2_*, and from the *bc*_1_ complex with already loaded quinols 16, altogether 60. These electrons can contribute to forming at most 30 quinols that become available at the *bc*_1_ complex for formation of the proton gradient. As the reviewer points out, it is not possible to explain the free energy buffer capacity of the chromatophore relevant for changes in light intensity from the present model without additional quinols.

However, in an actual chromatophore, the mechanism with which the redox state of the Q-pool is regulated involves also exchange of electrons with the succinate/fumarate pool in the cell's cytoplasm via the enzyme succinate dehydrogenase; additionally NADH dehydrogenase and cytochrome c oxidase facilitate electron exchange reactions with the Q-pool. The contribution of these reactions to the redox regulation of the Q-pool is currently not well understood and the corresponding enzymes are not resolved in the current structural model of the chromatophore shown in Figure 1. Therefore, in the current description of chromatophore processes the redox state of the Q-pool is accounted for, according to [Disp-formula equ14 equ15 equ16 equ17 equ18], in an heuristic way. For an explanation, an additional schematic figure (Figure 5) is added along with corresponding text (section 4.1, last paragraph) in the Materials and methods to introduce the missing enzymes. The role of these enzymes are discussed in: Introduction, fifth and sixth paragraphs; section 2.2, fifth paragraph; Discussion, fourth paragraph; section 4.1, last paragraph; Figure 5; subsection “Stage I: Light absorption, excitation energy transfer, and quinol formation”, eleventh paragraph; subsection “Stage II: Diffusion of charge carriers and estimate of cycling time τRC(I)”, fifth paragraph. This issue is further discussed in a note following response #1.4 below.

*2) Evolution did not act at the level of individual chromatophores but on the survival fitness of the intact organism.*

Of course, we agree with the reviewer and apologize for the “loose” language used in our original manuscript. As the reviewer states, evolution acts on the reproductive fitness of the entire organism. Yet, increased survival fitness for an organism frequently results from increased efficiency of individual cellular components (c.f. citations in Introduction, third paragraph; section 2.2, last paragraph), even though the direct relation between component efficiency and reproductive fitness is usually difficult to establish. This distinction is clarified in the text at the aforementioned paragraphs (Introduction, third paragraph; section 2.2, first paragraph).

*3) The authors use efficiency values without specifying exactly what they mean. I presume that they have had a certain numerical value for "delta-Gp" in mind (i.e. the standard free energy of hydrolysis of ATP), and then calculate efficiency as the ratio of the caloric value of the photons over the free energy of the corresponding ATP hydrolysis. This should however be made explicit. Furthermore, they now and then shift back and forth from oxygenic to anoxygenic photosynthesis (e.g. Discussion, first paragraph), which is not proper because in oxygenic photosynthesis the majority of the energy conservation goes into redox energy (i.e. NADPH). Estimates that anoxygenic photosynthesis has a free-energy efficiency in the order of 10% have already been available for a long time (Hellingwerf, K. J., Crielaard, W., & Westerhoff, H. V. (1993). Comparison of retinal-based and chlorophyll-based photosynthesis: A biothermokinetic description of photochemical reaction centers. Modern trends in biothermokinetics (pp. 45-52) Springer).*

The various definitions of energy conversion efficiency are compared and discussed now in: Materials and methods, first paragraph; subsection “Stage III: ATP synthesis”, second paragraph, particularly in the paragraph following the definition of overall conversion efficiency η _ATP in [Disp-formula equ20].

In following the reviewer's suggestion, we have removed the misleading comparison to plant photosynthesis from Discussion.

We also cite now the work of (Hellingwerf, 1993) that has provided an early estimate for a maximal value for the conversion efficiency of photosynthesis in *Rba. sphaeroides* (section 2.1, third paragraph). The main point of our study is not only to establish the efficiency value, but (i) to demonstrate that the efficiency value can be achieved on account of the overall molecular structure of the chromatophore; (ii) to discuss the dependence of ATP production rate and efficiency on light conditions; (iii) to determine the degree of optimization for energy conversion as a function of stoichiometry.

*4) Beyond the LH/RC itself, nowhere in the modelling has any leak-reaction taken into account. Two examples that would have been very relevant to take into account are: (a) the short-cut in electron transfer in the cytochrome b/c1 complex that may occur if the second electron of quinol is also directly transferred to the Rieske protein and (b) passive proton leakage through the chromatophore membrane (which may be very much dependent of the protein composition of the vesicles, due to curvature effects). Related to this is the fact that the authors seem to suggest that proton transport merely leads to a pH change (see discussion on acidification), whereas presumably the energetic coupling between the proton pumping system and the ATP synthase is through a transmembrane electrical potential gradient.*

Leak reactions mentioned by the reviewer must inevitably exist, but in the absence of sufficient data we do not account for them in the current model. This omission does not alter the primary conclusions that deal with the structure-function relationship of the chromatophore vesicle given the structurally resolved components (LH2, LH1, RC, *bc*_1_, ATP synthase) and cyt *c_2_*. For the description of steady-state kinetics based on rate-limiting effects in the system, it is not necessarily beneficial or informative to try to account for every subprocess in the system. Nonetheless, we discuss now for the benefit of the reader the presence of leak reactions as the reviewer suggests (Introduction, sixth paragraph; Discussion, fifth paragraph).

Additionally, we also stress once more that a proton gradient involves an electrical potential difference. However, referring throughout the text to “proton gradient and electrical potential difference” or the like makes the reading of the text cumbersome; readers will grasp what is meant, if after a first more complete description we refer in the following part of the text solely to "proton gradient".

Rerouting of electrons in the *bc*_1_ complex, mentioned by the referee as a leakage, is actually a more complex process leading to unwanted redox products, for example into the transformation of oxygen to superoxide, that needs to be controlled, e.g., through superoxide dismutase, and brings about more problems than just energy leakage (Rottenberg et al., J. Biol. Chem., 2009; 284: 19203-19210). Such rerouting is clearly beyond the scope of the present study that benefits from avoiding the discussion of numerous side reactions and keeping the focus on processes arising through the structurally resolved photosynthetic components of the chromatophore.

Since more than one reviewer brought up the possible inclusion of peripheral cellular processes in the present model of chromatophore energy conversion, we would like to note here that we consider the simplifications due to ignoring such processes a benefit of our model, rather than a detriment. The present focus on photosynthetic processes, grounded in known structural detail, leads to a clear interpretation. Nevertheless, the peripheral processes coupled to various trans-membrane enzymes are now listed for the reader in several portions of the text, including in a new figure (Figure 5). Specifically, the processes of flagellar motility, NAD(P)H synthesis, coupling to the succinate/fumarate pool, as well as leak-reactions are not considered. Of these processes, coupling to flagellar motility probably constitutes the largest consumption channel when active. Furthermore, a steady-state model is considered so as not to explicitly simulate the diffusion of quinone/quinol, cytochrome *c_2_*, and protons, but rather representing the respective concentrations through their averages. Simulation efforts are beginning to address these diffusion processes, but the timescales currently reachable (~150 ns) are insufficient to account for the diffusive regime (Chandler, et al., 2014. Light harvesting by lamellar chromatophores in *Rhodospirillum photometricum*. Biophys. J., 106:2503). The steady-state approach permits us to successfully account for the primary function of the chromatophore, determining ATP synthesis rates, overall energy conversion efficiency, and saturation of light harvesting capacity, as depicted in Figure 2 and Figure 3. To address reviewers' concerns for the benefit of the reader, we explicitly discuss, as mentioned already in response #1.1, the missing components of the chromatophore energy conversion absent from the current model in Introduction, fifth and sixth paragraphs; section 2.2, fifth paragraph; Discussion, fourth paragraph; section 4.1, sixth paragraph; subsection “Stage I: Light absorption, excitation energy transfer, and quinol formation”, eleventh paragraph; subsection “Stage II: Diffusion of charge carriers and estimate of cycling time τRC(I)”, fifth paragraph, as well as in an additional figure (Figure 5).

5) What precisely does the atomistic description of the light harvesting and trapping process contribute to the calculation of the efficiency? Could the energy transfer and trapping process not be captured by a single effective rate constant that is a function of the fraction of open and closed traps and the ratio of LH2 to RC-LH1. See many earlier papers on the subject: Vredenberg and Duysens, Duysens CIBA foundation, Den Hollander et al., Trissl et al., Joliot cs etc.

The referee brings up an important point: what value does structural cell biology bring to the understanding of cells; did past knowledge, like that achieved over five decades ago by Vredenberg and Duysens and mentioned by the referee (Nature 197: 355-357, 1963; doi:10.1038/197355a0) not suffice today? In general, it seems to be widely accepted that one re-interprets biological knowledge in terms of the more recently discovered atomic level structure of cellular components as they provide a further resolved and more fundamental understanding of cell biology. So, the authors feel well justified in describing photosynthesis, including light harvesting, in the present case on the basis of the atomic level structure of the apparatus established in recent work. They also are convinced that given the fundamental role of light harvesting, the great advances in structural biology and in finite temperature quantum physics of fluorescent systems, it seems a natural goal to develop from ground up the kinetics of the light harvesting apparatus, rather than employ heuristic descriptions from a period when structural details of light harvesting proteins and their arrangement were not known. Nevertheless, we gladly follow the reviewer's suggestion and cite some of the original discoveries now also directly (Introduction, ninth paragraph; section 2.1, sixth paragraph). The authors are grateful to the reviewer for the insight in linking the earlier work of the Dutch photosynthesis masters and the modern highly resolved picture of photosynthetic apparatus structure and function.

*Reviewer #2:*

The manuscript by Sener and co-workers presents a model of a Rhodobacter sphaeroides chromatophore. The model describes how the illumination of a chromatophore suspension results in ATP production. The ultimate goal of the modeling is to understand how the ATP yield is coupled with the intensity of light. The main virtue of the model is that it couples the quantum physics of light absorption/procession with the chemistry of ATP synthesis. The model gives reasonable values of ATP yield in response to low light, which is physiological for these bacteria. Development of such a global model, which covers events over a time span of 14 orders of magnitude, required certain simplifications, which is understandable. Unfortunately, from the opinion of this reviewer, the simplifications preclude from making physiologically relevant conclusions from the modeling, at least, at the given stage.

*Still the model itself is a great achievement and the manuscript deserves publication albeit after certain amendments.*

1) Since the physiological relevance of the results obtained is questionable, the main result of this study seems to be the model itself. Therefore, the description of the model should be moved from Materials and methods into the Results. Otherwise the manuscript is incomprehensive, taking into account that the format of eLife implies Materials and methods after Results. The Materials and methods section should then contain only technical/purely methodical points; supplementary materials, at least partly, could be also moved into Materials and methods.

We thank the reviewer for the suggestion to moving material of our text between Results, Materials and methods, and supplementary materials. Though this could make for better reading in case of readers following the text each time from the beginning to end, readers are familiar today with the Results/Materials and methods/supplementary materials division of articles in scientific journals like *eLife* and know to jump between these sections for information of methodological details. We decided for the present division since a reader can find the information of the actual conclusions of our study more easily in the present format. The reader is also asked now to read the Materials and methods section prior to proceeding with the Results section (Results, first paragraph; Materials and methods, first paragraph).

We also like to emphasize that the chromatophore structure, the ‘model’, is not the only result of our study; the other key result is the structure-function relationship of the whole chromatophore. This relationship links the overall structure of the chromatophore to ATP synthesis rate and energy conversion efficiency.

2) The current simplified model lacks two important modules, which, hopefully, would be added in the future. These are the protonmotive force and reversible NADH dehydrogenase. The authors admit the absence of these modules, but did not consider the consequences of their absence in full extent. This should be done in the Discussion.

The particular short-comings of the current model, including, in particular, the missing NADH dehydrogenase component, are now discussed at greater length in the text, as mentioned above following comments by reviewer #1, in (Introduction, fifth and sixth paragraphs; section 2.2, fifth paragraph; Discussion, fourth paragraph; section 4.1, sixth paragraph; subsection “Stage I: Light absorption, excitation energy transfer, and quinol formation”, eleventh paragraph; subsection “Stage II: Diffusion of charge carriers and estimate of cycling time τRC(I)”, fifth paragraph) as well as in an additional figure (Figure 5).

The proton gradient is left out of the analysis as proton migration from the *bc*_1_ complex to ATP synthase is not rate determining in chromatophore vesicles; we comment on the expected, fast proton migration rate in chromatophores in the text (section 4.2, fourth paragraph).

*3) In the absence of these two modules, the authors conclude that "energy conversion in the chromatophore appears to be rate-limited primarily by cytbc_1_" and that "cytbc_1_-limited kinetics prevents an overproduction of protons at sustained high-light conditions,…thereby protecting the chromatophore against overacidification of its interior and assuring vesicle integrity at high-light conditions".*

*In real phototropic bacteria, the situation is more complex.*

As mentioned for issue #2.2, we now discuss at length in the text the complexity of the coupling between the chromatophore energy conversion processes and cytoplasmic processes in the cell facilitated through NADH dehydrogenase, succinate dehydrogenase and other enzymes (Introduction, fifth and sixth paragraphs; section 2.2, fifth paragraph; Discussion, fourth paragraph; section 4.1, sixth paragraph; subsection “Stage I: Light absorption, excitation energy transfer, and quinol formation”, eleventh paragraph; subsection “Stage II: Diffusion of charge carriers and estimate of cycling time τRC(I)”, fifth paragraph; newly added figure, Figure 5). In particular, the rate-limiting role of *bc*_1_ and its relation to missing modules of the chromatophore is emphasized in the Discussion section (fourth paragraph; section 4.1, last paragraph).

Obviously, the function of a photosynthetic apparatus, such as bacterial chromatophore or a chloroplast thylakoid, is to produce ATP and NAD(P)H. In thylakoids, NAD(P)H molecules are produced via direct reduction of NAD(P)+ by the Photosystem I. In Rb. sphaerides, the redox potential of the RC is too high for NAD(P)H generation. Therefore, NAD(P)H is produced via reversing the NAD(P)H dehydrogenase. The reversion requires (1) high protonmotive force and (2) reduced ubiquinol pool. Therefore, the overreduction of the ubiquinol pool in Rb. sphaeroides is a precondition for the NAD(P)H production; it should increase the overall efficiency of the system. Furthermore, the overacidification of the interior is prevented not by the overreduction of the ubiquinone pool, but by the high sensitivity of the cytbc_1_ to the membrane potential. High voltage, needed to drive the NAD(P)+ reduction (see above), slows the turnover of the cytbc_1_ dramatically, essentially blocking its turnover at voltages above 150 mV. Hence at high, physiological values of protonmotive force the turnover of the cytbc_1_ is essentially limited by the proton discharge, specifically via the ATP synthase and the reverse NAD(P)H dehydrogenase. The operation of the cytbc_1_ close to equilibrium additionally increases the efficiency of energy conversion.

The reviewer raises great points about NAD(P)H production. We have taken the liberty to incorporate this comment in the text (Introduction: sixth paragraph; Results: section 2.2, fifth and sixth paragraphs; Discussion: fourth paragraph) and have added a corresponding Acknowledgment. As pointed out already in comments to referee #1 above, we emphasize the role of cytoplasmic redox reactions in the new text in all sections of the text and point to the important function of these reactions in regulating the redox state of the chromatophore Q-pool.

The reviewer admits that the data on synthesis of NAD(P)H in Rb. sphaeroides are too scarce to be used for verification of the model. The authors, accordingly, have tested their model by using more abundant data that were obtained with chromatophore suspensions in the absence of NAD(P)+, in experiments where the function of the RC, cytbc_1_ and/or ATP synthase were studied. And still, the discussion of physiological consequences from the given model should be truncated dramatically as misleading. The physiology could be discussed in further publications, after the membrane potential and NAD(P)H synthesis would be explicitly included in the model.

As pointed out above, we have added extensive discussions, based on reviewer comments, regarding the complexity of chromatophore energy conversion processes; the reader is made well aware of the presence of related cytoplasmic processes in the cell; and the interpretation about the buffering capacity in the system has been removed (see issue #1.1). Specifically, the following passages were modified: last sentence of the Abstract; Introduction, fifth paragraph; section 2.2, fourth and fifth paragraphs; Discussion, second and third paragraphs; section 4.2, fourth paragraph. We insist, however, that the consequences of the current structural-functional model be discussed in detail while informing the reader at length of the limitations of the current study.

Instead, the Discussion section should include comparison of the given model with the earlier models of chromatophore systems (e.g. from the group of Helms) and the approaches used.

The work of Geyer and Helms is already cited and discussed abundantly in the text, particularly, their conclusion that the primary rate-limiting step in the chromatophore stems from cyt*bc_1_*turnover. We add a further comparison in the Discussion section (fifth paragraph).

*Reviewer #3:*

*The authors present an important theoretical investigation of the performance and limitations of the photosynthetic chromatophore complex of purple bacteria. This light-harvesting system is of great interest as a model system that contains numerous characteristic properties inherent to all natural light-harvesting systems. This work is a continuation of a long line of first-rate papers published by the authors on this topic, is extremely well written, and merits publication in its current form.*

There appears to be a minor inconsistency in the units of intensity employed. In the second paragraph of section 4.2 the authors define light intensity, I, in units of [photons absorbed per second], whereas light intensity is often defined in units of [photons absorbed per meter squared per second]. As defined, this intensity is in fact a power, which is also called a 'radiant flux' in radiometry. This in contrast with the more conventional definition of intensity in the caption of Figure 2, where units of [Watts per meter squared] are used. Clarification of these different definitions would be useful.

We thank the reviewer for pointing out a potential source of confusion (and also for the related next comment). The text has been altered to clarify the definition of I (photons/s absorbed by entire vesicle) as discussed in (section 4.2, second and third paragraphs).

In the third paragraph of section 4.2, the authors switch to defining 'I' as 'illumination,' which may further confuse the reader. Ideally 'radiant flux' or 'absorbed light power' or similar terminology would be employed throughout, or alternatively 'illumination' would be defined.

Following the reviewer's suggestion, we refer to the quantity 'I' in the text as 'absorbed light power'.

The explanation provided in the fourth and fifth paragraphs of section 2.2 for the suboptimal (nB, nRC) values in low-light adapted vesicles is appealing, but it does not appear to be supported by calculations. Despite the fact that the model only applies in steady-state, is it possible to perform a simple simulation to support this interpretation, even qualitatively?

We are grateful to the reviewer for pointing out this issue, also discussed in response to issue #1.1. We concede that our current steady-state model is not able to generate quantitative conclusions about the response of the system to fluctuating light conditions and the corresponding energy storage & buffering effect for intermittent dark periods. The related discussion (section 2.2, fourth and fifth paragraphs) is altered to reflect this point. Please see our extended response to reviewer 1, issue #1. However, a non-steady state simulation is currently prohibitive computationally, since explicit spatial dynamics of charge carriers and quinones/quinols in the chromatophore need to be modeled, which goes beyond the technically feasible limit.

*The choice of initial condition assumed in [Disp-formula equ9] may possibly be over-simplifying because each complex (LH2, LH1 and RC) does not necessarily have the same absorption spectrum, since at a given wavelength, the probability of absorption of a photon is not the same for each complex. In this scenario, the initial condition should then be wavelength dependent. While the authors state that the quantum yield is not strongly dependent on the choice of P(0), a specific example to demonstrate this would be helpful.*

A discussion is added after [Disp-formula equ8 equ9], defining the quantum yield q in terms of the initial state P(0) and explaining that variations in P(0) affect *q* by only about 3%. The text states that this result was demonstrated in (Seneret al., 2007, PNAS, 104:15723) and, therefore, wavelength dependence of q is not further considered in the present study.

[Editors' note: further revisions were requested prior to acceptance, as described below.]

*On the basis of the referee reports on the revised manuscript, only relatively minor additional changes are required. Reviewer #2 is concerned that some issues have not been fully addressed in the revision. In the following I list the main points.*

*1) The reviewer maintains that "the model does not consider the production of NADH at all." Therefore, "no statements on the energy conversion efficiency could be done with the current model." The reviewer is concerned that the issue of NADH production and its impact on the model and its assessment are not discussed adequately, and advises "to read the primary literature on the mechanism of NADH production and provide references to this literature (currently they refer only to the paper of Klamt et al. where bacterial photosynthesis was modeled by using several dozen parameters that were more or less freely variable). The primary literature describes how the NADH-dehydrogenase makes NADH by driving the reverse (uphill) electron transfer from the ubiquinone pool to NAD+ with the help of membrane potential. This mechanism differs from the NAD(P)H production in green plants where NAD(P)+ is directly reduced by photoexcitated electrons in a reaction that is independent of membrane potential. The dependence of the light-driven NADH production on membrane potential, specific for Rhodobacter and other purple phototrophic bacteria, should have shaped the properties of their energy-converting machinery."*

Recommended action: Whereas I do not see the lack of NADH production as a fundamental flaw in the model (after all, all models are incomplete), this point should be made clearer to the reader. Conclusions drawn from the model should be qualified accordingly. In particular, in the discussion of the energy conversion efficiency, relevant limitations of the model should be stated.

We were surprised about the reviewer response, since the presence of NADH production as well as the need for modeling it in future studies was stated repeatedly in the revised manuscript, while also discussing why it cannot be accounted for in the current study due to the absence of the relevant proteins from current structural data. In particular, we stated that our model, through [Disp-formula equ15 equ16 equ17 equ18 equ19], describes the light intensity dependence of the quinone/quinol pool redox state that comes about through quinone-quinol conversion involving action of NADH dehydrogenase alongside other redox factors; thus, the influence of the redox factors are currently accounted for heuristically in the absence of structural data. We have now further emphasized this issue and the resulting shortcomings of the model in the main text at the following locations: Introduction, sixth paragraph; Discussion, fourth paragraph; section 4.1 last paragraph; section 4.2, last paragraph; subsection “Stage I: Light absorption, excitation energy transfer, and quinol formation”, ninth paragraph; subsection “Stage II: Diffusion of charge carriers and estimate of cycling time τRC(I)”, first paragraph.

*2) The reviewer also asks to point out clearly that "the experimental data that were used for testing the model were not obtained with chromatophores in vivo," but for "an artificial system – a suspension of chromatophores in a pH-buffer, in the presence of electron donors and redox mediators, but in the absence of physiological compounds such as NAD+/NADH (a Dutton/Crofts system)."*

Recommended action: Point out the distinction between the in-vitro experiments and the more complex in-vivo situation.

This distinction and the increased complexity of the in vivo situation is now emphasized in the text at: section 4.1, end of first paragraph.

*3) The reviewer also argues that the effect of a transmembrane proton gradient, which is not included explicitly in the model, should be discussed. The reviewer emphasizes that the issue is not "proton migration from the bc_1_ complex to ATP synthase" but protonmotive force.*

Recommended action: Discuss the issue of protonmotive force. Examine which conclusions would likely be impacted by the explicit treatment of PMF in the model, and qualify them accordingly. If none of the conclusions would be affected, state that the model is robust with respect to the neglect of explicit modeling of PMF.

We have added additional text in Discussion (sixth paragraph) and Methods (subsection “Stage I: Light absorption, excitation energy transfer, and quinol formation”, eleventh paragraph) to emphasize that the energy conversion model does not require the explicit treatment of proton translocation steps and is robust with respect to the neglect of explicit modeling of PMF in its prediction of ATP turnover and efficiency.

*4) The reviewer suggests adding a description of the model to the Results section.*

As already argued in our last submission, regarding the fourth issue we would rather keep the division between Results, Methods, and supplementary information in the present form.